# SITReg: Multi-resolution architecture for symmetric, inverse consistent, and topology preserving image registration

## Abstract

Deep learning has emerged as a strong alternative for classical iterative methods for deformable medical image registration, where the goal is to find a mapping between the coordinate systems of two images. Popular classical image registration methods enforce the useful inductive biases of symmetricity, inverse consistency, and topology preservation by construct. However, while many deep learning registration methods encourage these properties via loss functions, none of the methods enforces all of them by construct. Here, we propose a novel registration architecture based on extracting multi-resolution feature representations which is by construct symmetric, inverse consistent, and topology preserving. We also develop an implicit layer for memory efficient inversion of the deformation fields. Our method achieves state-of-the-art registration accuracy on two datasets.

## 1 Introduction

Deformable medical image registration aims at finding a mapping between coordinate systems of two images, called a *deformation*, to align them anatomically. Deep learning can be used to train a registration network which takes as input two images and outputs a deformation. We focus on unsupervised intra-modality registration without a ground-truth deformation and where images are of the same modality, applicable, e.g., when deforming brain MRI images from different patients to an atlas or analyzing a patient's breathing cycle from multiple images. To improve registration quality, various inductive biases have been assumed by previous methods: *inverse consistency*, *symmetry*, and *topology preservation* (Sotiras et al., 2013). Some of the most popular classical methods enforce these properties by construct (Ashburner, 2007; Avants et al., 2008). However, no such deep learning method exists, and we address this gap (see a detailed literature review in Appendix G). We start by defining the properties (further clarifications in Appendix M).

We define a *registration method* as a function $f$ that takes two images, say $x_A$ and $x_B$, and produces a deformation. Some methods can output the deformation in both directions, and we use subscripts

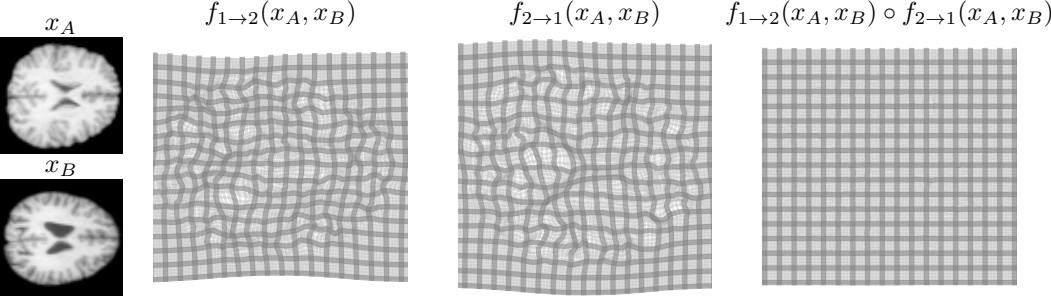

Figure 1: **Example deformation from the method.** *Left:* Forward deformation. *Middle:* Inverse deformation. *Right:* Composition of the forward and inverse deformations. Only one 2D slice is shown of the 3D deformation. The deformation is from the LPBA40 experiment. For more detailed visualization of a predicted deformation, see Figure 10 in Appendix J.

to indicate the direction. For example, $f_{1\rightarrow2}$ produces a deformation that aligns the image of the first argument to the image of the second argument. As a result, a registration method may predict up to four different deformations for any given input pair: $f_{1\rightarrow2}(x_A, x_B)$, $f_{2\rightarrow1}(x_A, x_B)$, $f_{1\rightarrow2}(x_B, x_A)$, and $f_{2\rightarrow1}(x_B, x_A)$. Some methods predict deformations in one direction only, resulting in two possible outputs: $f_{1\rightarrow2}(x_A, x_B)$ and $f_{1\rightarrow2}(x_B, x_A)$, in which case we might omit the subscript.

*Inverse consistent* registration methods ensure that $f_{1\rightarrow2}(x_A, x_B)$ is an accurate inverse of $f_{2\rightarrow1}(x_A, x_B)$, which we quantify using the *inverse consistency error*: $||f_{1\rightarrow2}(x_A, x_B) \circ f_{2\rightarrow1}(x_A, x_B) - \mathcal{I}||^2$, where $\circ$ is the composition operator and $\mathcal{I}$ is the identity deformation. Originally inverse consistency was achieved via variational losses (Christensen et al., 1995) but later algorithms were *inverse consistent by construct*, e.g., classical methods DARTEL (Ashburner, 2007) and SyN (Avants et al., 2008). However, due to a limited spatial resolution of the predicted deformations, even for these methods the inverse consistency error is not exactly zero. Some deep learning methods enforce inverse consistency via a penalty (Zhang, 2018; Kim et al., 2019; Estienne et al., 2021). A popular stationary velocity field (SVF) formulation (Arsigny et al., 2006) achieves inverse consistency by construct and has been used by many works, e.g., Dalca et al. (2018); Krebs et al. (2018; 2019); Niethammer et al. (2019); Shen et al. (2019a;b); Mok & Chung (2020a).

In *symmetric registration*, the registration outcome does not depend on the order of the inputs, i.e., $f_{1\rightarrow2}(x_A, x_B)$ equals $f_{2\rightarrow1}(x_B, x_A)$. Unlike with inverse consistency, $f_{1\rightarrow2}(x_A, x_B)$ can equal $f_{2\rightarrow1}(x_B, x_A)$ exactly (Avants et al., 2008; Estienne et al., 2021), which we call *symmetric by construct*. A related property, cycle consistency, can be assessed using *cycle consistency error* $||f(x_A, x_B) \circ f(x_B, x_A) - \mathcal{I}||^2$. It can be computed for any method since it does not require the method to predict deformations in both directions. If the method is symmetric by construct, inverse consistency error equals cycle consistency error. Some existing deep learning registration methods enforce cycle consistency via a penalty (Mahapatra & Ge, 2019; Gu et al., 2020; Zheng et al., 2021). The method by Estienne et al. (2021) is symmetric by construct but only for a single component of their multi-step formulation, and not inverse consistent by construct. Recently, parallel with and unrelated to us, Iglesias (2023); Greer et al. (2023) have proposed by construct symmetric and inverse consistent registration methods within the SVF framework, in a different way from us.

We define *topology preservation* of predicted deformations similarly to Christensen et al. (1995). From the real-world point of view this means the preservation of anatomical structures, preventing non-smooth changes. Mathematically we want the deformations to be homeomorphisms, i.e., invertible and continuous. In practice we want a deformation not to fold on top of itself which we measure by estimating the local Jacobian determinants of the predicted deformations, and checking whether they are positive. Most commonly in deep learning applications topology preservation is achieved using the diffeomorphic SVF formulation (Arsigny et al., 2006). It does not completely prevent the deformation from folding due to limited sampling resolution but such voxels are limited to very small percentage, which is sufficient in practice. Topology preservation can be encouraged with a specific loss, e.g., by penalizing negative determinants (Mok & Chung, 2020a).

Our main contributions can be summarized as follows:

- We propose a novel multi-resolution deep learning registration architecture which is by construct inverse consistent, symmetric, and preserves topology. The properties are fulfilled for the whole multi-resolution pipeline, not just separately for each resolution. Apart from the parallel work by (Greer et al., 2023), we are not aware of other multi-resolution deep learning registration methods which are by construct both symmetric and inverse consistent. For motivation of the multi-resolution approach, see Section 2.2.

- As a component in our architecture, we propose an *implicit* neural network layer, which we call *deformation inversion layer*, based on a well-known fixed point iteration formula (Chen et al., 2008) and recent advances in Deep Equilibrium models (Bai et al., 2019; Duvenaud et al., 2020). The layer allows memory efficient inversion of deformation fields.

- We show that the method achieves state-of-the-art results on two popular benchmark data sets in terms of registration accuracy and deformation regularity. The accuracy of the inverses generated by our method is also very good and similar to the s-o-t-a SVF framework.

We name the method *SITReg* after its symmetricity, inverse consistency and topology preservation properties.

## 2 BACKGROUND AND PRELIMINARIES

### 2.1 TOPOLOGY PRESERVING REGISTRATION

The LDDMM method by (Cao et al., 2005) is a classical registration method that can generate diffeomorphic deformations which preserve topology, but it has not been used much in deep learning due to computational cost. Instead, a simpler stationary velocity field (SVF) method (Arsigny et al., 2006) has been popular (Krebs et al., 2018; 2019; Niethammer et al., 2019; Shen et al., 2019a;b; Mok & Chung, 2020a). In SVF the final deformation is obtained by integrating a stationary velocity field over itself over a unit time, which results in a diffeomorphism. Another classical method by Choi & Lee (2000); Rueckert et al. (2006) generates invertible deformations by constraining each deformation to be diffeomorphic but small, and forming the final deformation as a composition of multiple small deformations. Since diffeomorphisms form a group under composition, the final deformation is diffeomorphic. This is close to a practical implementation of the SVF, where the velocity field is integrated by first scaling it down by a power of two and interpreting the result as a small deformation, which is then repeatedly composed with itself. The idea is hence similar: a composition of small deformations.

In this work we build topology preserving deformations using the same strategy, as a composition of small topology preserving deformations.

### 2.2 MULTI-RESOLUTION REGISTRATION

Multi-resolution registration methods learn the deformation by first estimating it in a low resolution and then incrementally improving it while increasing the resolution. For each resolution one feeds the input images deformed with the deformation learned thus far, and incrementally composes the full deformation. Since its introduction a few decades ago (Rueckert et al., 1999; Oliveira & Tavares, 2014), the approach has been used in the top-performing classical and deep learning registration methods (Avants et al., 2008; Klein et al., 2009; Mok & Chung, 2020b; 2021; Hering et al., 2022).

In this work we propose the first multi-resolution deep learning registration architecture that is by construct symmetric, inverse consistent, and topology preserving.

### 2.3 SYMMETRIC REGISTRATION FORMULATIONS

Symmetric registration does not assign moving or fixed identity to either image but instead considers them equally. A classical method called symmetric normalization (SyN, Avants et al., 2008) is a symmetric registration algorithm which learns two separate transformations: one for deforming the first image half-way toward the second image and the other for deforming the second image half-way toward the first image. The images are matched in the intermediate coordinates and the full deformation is obtained as a composition of the half-way deformations (one of which is inverted). The same idea has later been used by other methods such as the deep learning method SYMNet Mok & Chung (2020a). However, SYMNet does not guarantee symmetricity by construct (see Figure 6 in Appendix I).

We also use the idea of deforming the images half-way towards each other to achieve inverse consistency and symmetry throughout our multi-resolution architecture.

### 2.4 DEEP EQUILIBRIUM NETWORKS

Deep equilibrium networks use *implicit* fixed point iteration layers, which have emerged as an alternative to the common *explicit* layers (Bai et al., 2019; 2020; Duvenaud et al., 2020). Unlike explicit layers, which produce output via an exact sequence of operations, the output of an implicit layer is defined indirectly as a solution to a fixed point equation, which is specified using a fixed point mapping. In the simplest case the fixed point mapping takes two arguments, one of which is the input. For example, let $g : A \times B \to B$ be a fixed point mapping defining an implicit layer. Then, for a given input $a$, the output of the layer is the solution $z$ to equation

$$z = g(z, a). \tag{1}$$

This equation is called a fixed point equation and the solution is called a fixed point solution. If $g$ has suitable properties, the equation can be solved iteratively by starting with an initial guess and

repeatedly feeding the output as the next input to $g$. More advanced iteration methods have also been developed for solving fixed point equations, such as Anderson acceleration (Walker & Ni, 2011).

The main mathematical innovation related to deep equilibrium networks is that the derivative of an implicit layer w.r.t. its inputs can be calculated based solely on a fixed point solution, i.e., no intermediate iteration values need to be stored for back-propagation. Now, given some solution $(a_0, z_0)$, such that $z_0 = g(z_0, a_0)$, and assuming certain local invertibility properties for $g$, the implicit function theorem says that there exists a solution mapping in the neighborhood of $(a_0, z_0)$, which maps other inputs to their corresponding solutions. Let us denote the solution mapping as $z^*$. The solution mapping can be seen as the theoretical explicit layer corresponding to the implicit layer. To find the derivatives of the implicit layer we need to find the Jacobian of $z^*$ at point $a_0$ which can be obtained using implicit differentiation as

$$\partial z^*(a_0) = [I - \partial_1 g(z_0, a_0)]^{-1} \partial_0 g(z_0, a_0).$$

The vector-Jacobian product of $z^*$ needed for back-propagation can be calculated using another fixed point equation without fully computing the Jacobians, see, e.g., Duvenaud et al. (2020). Hence, both forward and backward passes of the implicit layer can be computed as a fixed point iteration.

We use these ideas to develop a neural network layer for inverting deformations based on the fixed point equation, following Chen et al. (2008). The layer is very memory efficient as only the fixed point solution needs to be stored for the backward pass.

## 3 METHODS

Let $n$ denote the dimensionality of the image, e.g., $n = 3$ for 3D medical images, and $k$ the number of channels, e.g., $k = 3$ for an RGB-image. The goal in deformable image registration is to find a mapping from $\mathbb{R}^n$ to $\mathbb{R}^n$, connecting the coordinate systems of two non-aligned images $x_A, x_B : \mathbb{R}^n \to \mathbb{R}^k$, called a deformation. Application of a deformation to an image can be mathematically represented as a (function) composition of the image and the deformation, denoted by $\circ$. Furthermore, in practice linear interpolation is used to represent images (and deformations) in continuous coordinates.

In this work the deformations are in practice stored as displacement fields with the same resolution as the registered images, that is, each pixel or voxel is associated with a displacement vector describing the coordinate difference between the original image and the deformed image (e.g. if $n = 3$, displacement field is tensor with shape $3 \times H \times W \times D$ where $H \times W \times D$ is the shape of the image). In our notation we equate the displacement fields with the corresponding coordinate mappings, and always use $\circ$ to denote the deformation operation (sometimes called warping).

In deep learning based image registration, we aim at learning a *neural network* $f$ that takes two images as input and outputs a deformation connecting the image coordinates. Specifically, in medical context $f$ should be such that $x_A \circ f(x_A, x_B)$ matches anatomically with $x_B$.

### 3.1 SYMMETRIC FORMULATION

As discussed in Section 1, we want our method to be symmetric. To achieve this, we define the network $f$ using another *auxiliary network* $u$ which also predicts deformations:

$$f(x_A, x_B) := u(x_A, x_B) \circ u(x_B, x_A)^{-1}. \tag{2}$$

As a result, it holds that $f(x_A, x_B) = f(x_B, x_A)^{-1}$ apart from errors introduced by the composition and inversion, resulting in a very low cycle-consistency error. An additional benefit is that $f(x_A, x_A)$ equals the identity transformation, again apart from numerical inaccuracies, which is a natural requirement for a registration method. Applying the formulation in Equation 2 naively would double the computational cost. To avoid this we encode features from the inputs separately before feeding them to the deformation extraction network in Equation 2. A similar approach has been used in recent registration methods (Estienne et al., 2021; Young et al., 2022). Denoting the feature extraction network by $h$, the modified formulation is

$$f(x_A, x_B) := u(h(x_A), h(x_B)) \circ u(h(x_B), h(x_A))^{-1}. \tag{3}$$

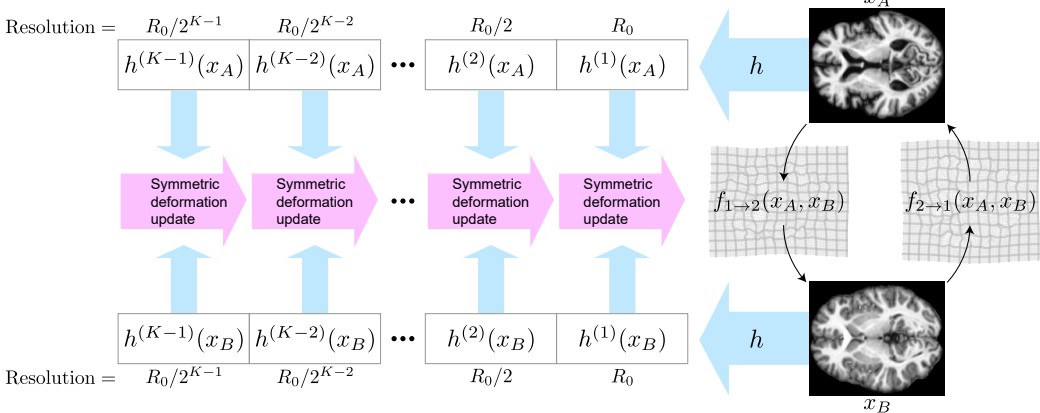

Figure 2: **Overview of the proposed architecture.** Multi-resolution features are first extracted from the inputs $x_A$ and $x_B$ using convolutional encoder $h$. Output deformations $f_{1\to2}(x_A, x_B)$ and $f_{2\to1}(x_A, x_B)$ are built recursively from the multi-resolution features using the symmetric deformation updates described in Section 3.2 and visualized in Figure 3. The architecture is symmetric and inverse consistent with respect to the inputs and the final deformation is obtained in both directions. The brain images are from the OASIS dataset (Marcus et al., 2007)

### 3.2 Multi-resolution architecture

As the overarching architecture, we propose a novel symmetric and inverse consistent multi-resolution coarse-to-fine approach. For motivation, see Section 2.2. Overview of the architecture is shown in Figure 2, and the prediction process is demonstrated visually in Figure 10 (Appendix J).

First, we extract image feature representations $h^{(k)}(x_A), h^{(k)}(x_B)$, at different resolutions $k \in \{0, \ldots, K-1\}$. Index $k = 0$ is the original resolution and increasing $k$ by one halves the spatial resolution. In practice $h$ is a ResNet (He et al., 2016) style convolutional network and features at each resolution are extracted sequentially from previous features. Starting from the lowest resolution $k = K - 1$, we recursively build the final deformation between the inputs using the extracted representations. To ensure symmetry, we build two deformations: one deforming the first image half-way towards the second image, and the other for deforming the second image half-way towards the first image (see Section 2.3). The full deformation is composed of these at the final stage. Let us denote the half-way deformations extracted at resolution $k$ as $d_{1\to1.5}^{(k)}$ and $d_{2\to1.5}^{(k)}$. Initially, at level $k = K$, these are identity deformations. Then, at each $k = K-1, \ldots, 0$, the half-way deformations are updated by composing them with a predicted update deformation. In detail, the update at level $k$ consists of three steps (visualized in Figure 3):

1. Deform the feature representations $h^{(k)}(x_A), h^{(k)}(x_B)$ of level $k$ towards each other by the half-way deformations from the previous level $k + 1$:

$$z_1^{(k)} := h^{(k)}(x_A) \circ d_{1\to1.5}^{(k+1)} \quad \text{and} \quad z_2^{(k)} := h^{(k)}(x_B) \circ d_{2\to1.5}^{(k+1)}. \tag{4}$$

2. Define an *update deformation* $\delta^{(k)}$, using the idea from Equation 3 and the half-way deformed feature representations $z_1^{(k)}$ and $z_2^{(k)}$:

$$\delta^{(k)} := u^{(k)}(z_1^{(k)}, z_2^{(k)}) \circ u^{(k)}(z_2^{(k)}, z_1^{(k)})^{-1}. \tag{5}$$

Here, $u^{(k)}$ is a trainable convolutional neural network predicting an invertible auxiliary deformation (details in Appendix A). The intuition here is that the symmetrically predicted update deformation $\delta^{(k)}$ should learn to adjust for whatever differences in the image features remain after deforming them half-way towards each other in Step 1 with deformations $d^{(k+1)}$ from the previous resolution.

3. Obtain the updated half-way deformation $d_{1\to1.5}^{(k)}$ by composing the earlier half-way deformation of level $k + 1$ with the update deformation $\delta^{(k)}$

$$d_{1\to1.5}^{(k)} := d_{1\to1.5}^{(k+1)} \circ \delta^{(k)}. \tag{6}$$

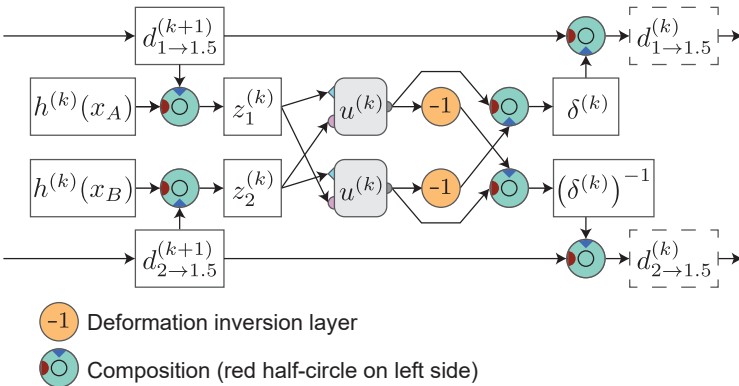

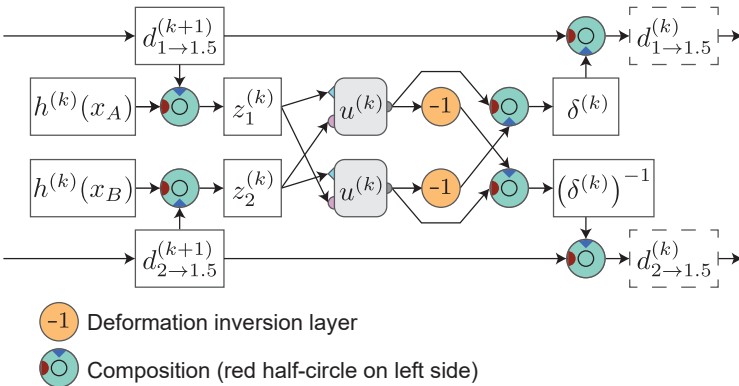 Deformation inversion layer

Composition (red half-circle on left side)

Figure 3: **Recursive multi-resolution deformation update.** The deformation update at resolution $k$, described in Section 3.2, takes as input the half-way deformations $d_{1 \to 1.5}^{(k+1)}$ and $d_{2 \to 1.5}^{(k+1)}$ from the previous resolution, and updates them through a composition with an update deformation $\delta^{(k)}$. The update deformation $\delta^{(k)}$ is calculated symmetrically from image features $z_1^{(k)}$ and $z_2^{(k)}$ (deformed mid-way towards each other with the previous half-way deformations) using a neural network $u^{(k)}$ according to Equation 5. The deformation inversion layer for inverting auxiliary deformations predicted by $u^{(k)}$ is described in Section 3.3.

For the other direction $d_{2 \to 1.5}^{(k)}$, we use the inverse of the deformation update $\left( \delta^{(k)} \right)^{-1}$ which can be obtained simply by reversing $z_1^{(k)}$ and $z_2^{(k)}$ in Equation 5 (see Figure 3):

$$d_{2 \to 1.5}^{(k)} = d_{2 \to 1.5}^{(k+1)} \circ \left( \delta^{(k)} \right)^{-1}. \tag{7}$$

The inverses $\left( d_{1 \to 1.5}^{(k)} \right)^{-1}$ and $\left( d_{2 \to 1.5}^{(k)} \right)^{-1}$ are updated similarly.

The full registration architecture is then defined by the functions $f_{1 \to 2}$ and $f_{2 \to 1}$ which compose the half-way deformations from stage $k = 0$:

$$f_{1 \to 2}(x_A, x_B) := d_{1 \to 1.5}^{(0)} \circ \left( d_{2 \to 1.5}^{(0)} \right)^{-1} \quad \text{and} \quad f_{2 \to 1}(x_A, x_B) := d_{2 \to 1.5}^{(0)} \circ \left( d_{1 \to 1.5}^{(0)} \right)^{-1}. \tag{8}$$

Note that $d_{1 \to 1.5}^{(0)}$, $d_{2 \to 1.5}^{(0)}$, and their inverses are functions of $x_A$ and $x_B$ through the features $h^{(k)}(x_A), h^{(k)}(x_B)$ in Equation 4, but the dependence is suppressed in the notation for clarity.

By using half-way deformations at each stage, we avoid the problem with full deformations of having to select either of the image coordinates to which to deform the feature representations of the next stage, breaking the symmetry of the architecture. Now we can instead deform the feature representations of both inputs by the symmetrically predicted half-way deformations, which ensures that the updated deformations after each stage are separately invariant to input order.

### 3.3 IMPLICIT DEFORMATION INVERSION LAYER

Implementing the architecture requires inverting deformations from $u^{(k)}$ in Equation 5. This could be done, e.g., with the SVF framework, but we propose an approach which requires storing $\approx 5$ times less data for the backward pass than the standard SVF. The memory saving is significant due to the high memory consumption of volumetric data, allowing larger images to be registered. During each forward pass $2 \times (K - 1)$ inversions are required. More details are provided in Appendix F.

As shown by Chen et al. (2008), deformations can be inverted in certain cases by a fixed point iteration. Consequently, we propose to use the deep equilibrium network framework from Section 2.4 for inverting deformations, and label the resulting layer *deformation inversion layer*. The fixed point equation proposed by Chen et al. (2008) is

$$g(z, a) := -(a - \mathcal{I}) \circ z + \mathcal{I},$$

where $a$ is the deformation to be inverted, $z$ is the candidate for the inverse of $a$, and $\mathcal{I}$ is the identity deformation. It is easy to see that substituting $a^{-1}$ for $z$, yields $a^{-1}$ as output. We use Anderson acceleration (Walker & Ni, 2011) for solving the fixed point equation and use the memory-effecient back-propagation (Bai et al., 2019; Duvenaud et al., 2020) strategy discussed in Section 2.4.

Lipschitz condition is sufficient for the fixed point algorithm to converge (Chen et al., 2008), and we ensure that the predicted deformations fulfill the condition (see Appendix A). The iteration converges well also in practice as shown in Appendix H.

### 3.4 Theoretical properties

**Theorem 3.1.** *The proposed architecture is inverse consistent by construct.*

**Theorem 3.2.** *The proposed architecture is symmetric by construct.*

**Theorem 3.3.** *Assume that $u^{(k)}$ is defined as described in Appendix A. Then the proposed architecture is topology preserving.*

*Proof.* Appendix D, including discussion on numerical errors caused by limited sampling resolution.

### 3.5 Training and implementation

We train the model in an unsupervised end-to-end manner similarly to most other unsupervised registration methods, by using similarity and deformation regularization losses. The similarity loss encourages deformed images to be similar to the target images, and the regularity loss encourages desirable properties, such as smoothness, on the predicted deformations. For similarity we use local normalized cross-correlation with window width 7 and for regularization we use $L^2$ penalty on the gradients of the displacement fields, identically to VoxelMorph (Balakrishnan et al., 2019). We apply the losses in both directions to maintain symmetry. One could apply the losses in the intermediate coordinates and avoid building the full deformations during training. The final loss is:

$$\mathcal{L} = \text{NCC}(x_A \circ d_{1 \to 2},\ x_B) + \text{NCC}(x_A,\ x_B \circ d_{2 \to 1}) + \lambda \times [\text{Grad}(d_{1 \to 2}) + \text{Grad}(d_{2 \to 1})],\ (9)$$

where $d_{1 \to 2} := f_{1 \to 2}(x_A, x_B)$, $d_{2 \to 1} := f_{2 \to 1}(x_A, x_B)$, NCC the local normalized cross-correlation loss, Grad the $L^2$ penalty on the gradients of the displacement fields, and $\lambda$ is the regularization weight. For details on hyperparameter selection, see Appendix B. Our implementation is in PyTorch (Paszke et al., 2019). Code is included in supplementary materials, allowing replication. Evaluation methods and preprocessing done by us, see Section 4, are included. The repository will be published upon acceptance.

### 3.6 Inference

We consider two variants: **Standard**: The final deformation is formed by iteratively resampling at each image resolution (common approach). **Complete**: All individual deformations (outputs of $u^{(k)}$) are stored in memory and the final deformation is their true composition. The latter is included only to demonstrate that the deformation is everywhere invertible (no negative determinants) without numerical sampling errors, but the first one is used unless stated otherwise, and perfect invertibility is not necessary in practice. Due to limited sampling resolution even the existing "diffeomorphic" registration frameworks such as SVF do not usually achieve perfect invertibility.

## 4 Experimental setup

**Datasets:** We use two subject-to-subject registration datasets: *OASIS* brains dataset with 414 T1-weighted brain MRI images (Marcus et al., 2007) as pre-processed for Learn2Reg challenge (Hoopes et al., 2021; Hering et al., 2022) [1] , and *LPBA40* dataset from University of California Laboratory of Neuro Imaging (USC LONI) with 40 brain MRI images (Shattuck et al., 2008) [2] . Preprocessing for both datasets includes bias field correction, normalization, and cropping. For OASIS

---

[1]https://www.oasis-brains.org/#access
[2]https://resource.loni.usc.edu/resources/atlases/license-agreement/

Table 1: **Results, OASIS dataset.** Mean and standard deviation of each metric are computed on the test set. The percentage of folding voxels ($\%$ of $|J_\phi|_{\leq 0}$) from the complete SITReg version is shown in blue, other results are with the standard version (see Section 3.6). VoxelMorph and cLapIRN do not predict inverse deformations and hence the inverse-consistency error is not shown. Determinant standard deviation and the consistency metrics are omitted for the SITReg (raw data) since they are not comparable with others (omitted values were $1.0e{-}3(3.9e{-}4)/0(0)$, $0.18(0.028)$, $1.2e{-}3(2.7e{-}4)$, and $1.2e{-}3(2.7e{-}4)$).

| Model | Accuracy | | Deformation regularity | | Consistency | |
|---|---|---|---|---|---|---|
| | Dice $\uparrow$ | HD95 $\downarrow$ | $\%$ of $|J_\phi|_{\leq 0} \downarrow$ | $\text{std}(|J_\phi|) \downarrow$ | Cycle $\downarrow$ | Inverse $\downarrow$ |
| SYMNet (original) | 0.788(0.029) | 2.15(0.57) | **1.5e−3**(4.3e−4) | **0.44**(0.039) | 3.0e−1(2.9e−2) | **3.5e−3**(4.2e−4) |
| SYMNet (simple) | 0.787(0.029) | 2.17(0.58) | 1.5e−2(3.1e−3) | 0.46(0.045) | 2.8e−1(2.8e−2) | 5.2e−3(8.4e−4) |
| VoxelMorph | 0.803(0.031) | 2.08(0.57) | 1.4e−1(9.4e−2) | 0.49(0.032) | 4.5e−1(5.3e−2) | - |
| cLapIRN | 0.812(0.027) | 1.93(0.50) | 1.1e0(2.1e−1) | 0.55(0.032) | 1.2e0(1.6e−1) | - |
| SITReg | **0.818**(0.025)* | 1.84(0.45)* | 8.1e−3(1.6e−3)/**0**(0) | 0.45(0.038) | **5.5e−3**(6.9e−4) | 5.5e−3(6.9e−4) |
| SITReg (raw data) | 0.813(0.023)* | **1.80**(0.52)* | - | - | - | - |

$^*$ Statistically significant ($p < 0.05$) improvement compared to the baselines, for details see Appendix L.

dataset we use affinely pre-aligned images and for LPBA40 dataset we use rigidly pre-aligned images. Additionally we train our model without any pre-alignment on OASIS data (*OASIS raw*) to test our method with larger initial displacements. Voxel sizes of the affinely aligned and raw datasets are the same but volume sizes differ. Details of the split into training, validation, and test sets, and cropping and resolution can be found in Appendix K.

**Evaluation metrics:** We evaluate *registration accuracy* using segmentations of brain structures included in the datasets: (35 structures for OASIS and 56 for LPBA40), and two metrics: Dice score (Dice) and 95% quantile of the Hausdorff distances (HD95), similarly to Learn2Reg challenge (Hering et al., 2022). Dice score measures the overlap of the segmentations of source images deformed by the method and the segmentations of target images, and HD95 measures the distance between the surfaces of the segmentations. However, comparing methods only based on the overlap of anatomic regions is insufficient (Pluim et al., 2016; Rohlfing, 2011), and hence also *deformation regularity* should be measured, for which we use conventional metrics based on the local Jacobian determinants at $10^6$ sampled locations in each volume. The local derivatives were estimated via small perturbations of $10^{-7}$ voxels. We measure topology preservation as the proportion of the locations with a negative determinant ($\%$ of $|J_\phi|_{\leq 0}$), and deformation smoothness as the standard deviation of the determinant ($\text{std}(|J_\phi|)$). Additionally we report inverse and cycle *consistency* errors, see Section 1.

**Baselines:** We compare against *VoxelMorph* (Balakrishnan et al., 2019), *SYMNet* (Mok & Chung, 2020a), and conditional LapIRN (*cLapIRN*) (Mok & Chung, 2020b; 2021). VoxelMorph is a standard baseline in deep learning based unsupervised registration. With SYMNet we are interested in how well our method preserves topology and how accurate the generated inverse deformations are compared to the SVF based methods. Additionally, since SYMNet is symmetric from the loss point of view, it is interesting to see how symmetric predictions it produces in practice. cLapIRN was the best method on OASIS dataset in Learn2Reg 2021 challenge (Hering et al., 2022). We used the official implementations[3][4][5] adjusted to our datasets. SYMNet uses anti-folding loss to penalize negative determinant. Since this loss is a separate component that could be easily used with any method, we also train SYMNet without it, denoted *SYMNet (simple)*. This provides a comparison on how well the vanilla SVF framework can generate invertible deformations in comparison to our method. For details on hyperparameter selection for baseline models, see Appendix C.

## 5 RESULTS

Evaluation results are in Tables 1 and 2, and additional visualizations in Appendix I. The assessment of registration performance should not be based on a single metric, e.g., accuracy, but instead on the

---

[3]https://github.com/voxelmorph/voxelmorph

[4]https://github.com/cwmok/Fast-Symmetric-Diffeomorphic-Image-Registration-with-Convolutional-Neural-Networks

[5]https://github.com/cwmok/Conditional_LapIRN/

Table 2: **Results, LPBA40 dataset.** The results are interpreted similarly to Table 1.

| Model | Accuracy | | Deformation regularity | | Consistency | |
|---|---|---|---|---|---|---|
| | Dice $\uparrow$ | HD95 $\downarrow$ | % of $\|J_\phi\|_{\leq 0} \downarrow$ | std($\|J_\phi\|$) $\downarrow$ | Cycle $\downarrow$ | Inverse $\downarrow$ |
| SYMNet (original) | 0.669(0.033) | 6.79(0.70) | **1.1e−3**(4.6e−4) | 0.35(0.050) | 2.7e−1(6.1e−2) | **2.1e−3**(4.3e−4) |
| SYMNet (simple) | 0.664(0.034) | 6.88(0.73) | 4.7e−3(1.6e−3) | 0.37(0.053) | 2.8e−1(5.8e−2) | 2.9e−3(6.7e−4) |
| VoxelMorph | 0.676(0.032) | 6.72(0.68) | 2.2e−1(2.1e−1) | 0.35(0.040) | 3.1e−1(1.1e−1) | - |
| cLapIRN | 0.714(0.019) | 5.93(0.43) | 8.4e−2(2.9e−2) | **0.27**(0.020) | 5.6e−1(1.8e−1) | - |
| SITReg | **0.720**(0.017)* | **5.88**(0.43) | 2.4e−3(6.4e−4)/**0**(0) | 0.31(0.032) | **2.6e−3**(4.2e−4)* | 2.6e−3(4.2e−4) |

* Statistically significant ($p < 0.05$) improvement compared to the baselines, for details see Appendix L.

Table 3: **Computational efficiency, OASIS dataset.** Mean and standard deviation are shown. Inference time and memory usage were measured on NVIDIA GeForce RTX 3090. Images in the raw dataset without pre-alignment have 3.8 times more voxels, significantly increasing the inference time and memory usage.

| Model | Inference Time (s) $\downarrow$ | Inference Memory (GB) $\downarrow$ | # parameters (M) $\downarrow$ |
|---|---|---|---|
| SYMNet (original) | **0.095**(0.00052) | **1.9** | **0.9** |
| SYMNet (simple) | 0.096(0.00044) | **1.9** | **0.9** |
| VoxelMorph | 0.16(0.0010) | 5.6 | 1.3 |
| cLapIRN | 0.10(0.00052) | 4.1 | 1.2 |
| SITReg | 0.37(0.0057) | 3.4 | 1.2 |
| SITReg (raw data) | 1.9(0.022) | 12.1 | 2.5 |

overall performance across different metrics, similarly to the Learn2reg challenge (Hering et al., 2022). This is because of the trade-off invoked by the regularization hyperparameter (in our case $\lambda$) between the accuracy and regularization metrics (an extreme example is presented by Rohlfing (2011)), which can change the ranking of the methods for different metrics. In such an overall comparison, our method clearly outperforms all baselines on both datasets, as summarized below:

- **VoxelMorph:** Our method outperforms it on every metric.

- **SYMNet:** While *SYMNet (original)* has slightly fewer folding voxels (compared to our standard inference variant) and slightly better inverse consistency, our method has a significantly better dice score. By increasing regularization one could easily make our model to have better regularity while maintaining significantly better dice score than SYMNet. Indeed, based on validation set results in Table 8 in Appendix B, by setting $\lambda = 2$, our model would be superior on every metric compared to SYMNet. In other words, our method has significantly better overall performance.

- **cLapIRN:** While in terms of tissue overlap metrics our method is only slightly better than cLapIRN, our method has significantly better deformation regularity in terms of folding voxels, and also significantly better cycle consistency.

A comparison of the methods' efficiencies is in Table 3. Inference time of our method is slightly larger than that of the compared methods, but unlike VoxelMorph and cLapIRN, it produces deformations in both directions immediately. Also, half a second runtime is still very fast and restrictive only in the most time-critical use cases. In terms of memory usage our method is very competitive.

## 6 CONCLUSIONS

We proposed a novel image registration architecture inbuilt with the desirable inductive biases of symmetry, inverse consistency, and topology preservation. The multi-resolution formulation was capable of accurately registering images even with large intial misalignments. As part of our method, we developed a new neural network component *deformation inversion layer*. The model is easily end-to-end trainable and does not require tedious multi-stage training strategies. In the experiments the method demonstrated state-of-the-art registration performance. The main limitation is somewhat heavier computational cost than other methods.

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

## A TOPOLOGY PRESERVING DEFORMATION PREDICTION NETWORKS

Each $u^{(k)}$ predicts a deformation based on the features $z_1^{(k)}$ and $z_2^{(k)}$ and we define the networks $u^{(k)}$ as CNNs predicting cubic spline control point grid in the resolution of the features $z_1^{(k)}$ and $z_2^{(k)}$. The use of cubic spline control point grid for defining deformations is a well-known strategy in image registration, see e.g. (Rueckert et al., 2006; De Vos et al., 2019).

However, deformations generated by $u^{(k)}$ have to be invertible to ensure the topology preservation property, and particularly invertible by the deformation inversion layer. To ensure that, we limit the control point absolute values below a hard contraint $\gamma^{(k)}$ using a scaled Tanh function.

In more detail, each $u^{(k)}$ then consists of the following sequential steps:

1. Concatenation of the two inputs, $z_1^{(k)}$ and $z_2^{(k)}$, along the channel dimension. Before concatenation we reparametrize the features as $z_1^{(k)} - z_2^{(k)}$ and $z_1^{(k)} + z_2^{(k)}$ as suggested by Young et al. (2022).

2. Two convolutions with kernel of spatial size 3 with ReLU activation after each of the convolutions.

3. Convolution with kernel of spatial size 1 and number of dimensions as output channels.

4. $\gamma^{(k)} \times$ Tanh function

5. Cubic spline upsampling to the image resolution by interpreting the output of the step 4 as cubic spline control point grid, similarly to e.g. De Vos et al. (2019). Cubic spline upsampling can be effeciently implemented as one dimensional transposed convolutions along each axis.

As shown in Appendix E, optimal upper bound for $\gamma^{(k)}$ ensuring invertibility can be obtained by the formula $\gamma^{(k)} < \frac{1}{K_n^k}$ where

$$K_n^{(k)} := \max_{x \in X} \sum_{\alpha \in \mathbb{Z}^n} \left| \sum_{j=1}^{n} \frac{B(x_j + \frac{1}{2^k} - \alpha_j) - B(x_j - \alpha_j)}{1/2^k} \prod_{i \in N \setminus \{j\}} B(x_i - \alpha_i) \right|, \qquad (10)$$

$n$ is the dimensionality of the images (in this work $n = 3$), $X := \{\frac{1}{2} + \frac{1}{2^{k+1}} + \frac{i}{2^k} \mid i \in \mathbb{Z}\}^n \cap [0,1]^n$ are the relative sampling positions used in cubic spline upsampling (imlementation detail), and $B$ is a centered cardinal cubic B-spline (symmetric function with finite support). In practice we define $\gamma^{(k)} := 0.99 \times \frac{1}{K_n^{(k)}}$.

The formula can be evaluated exactly for dimensions $n = 2, 3$ for a reasonable number of resolution levels (for concrete values, see Table 4). Note that the outer sum over $\mathbb{Z}^n$ is finite since $B$ has a finite support and hence only a finite number of terms are non-zero.

Table 4: Values of $K_2$ and $K_3$ for different sampling rates with respect to the control point grid. The bound for Sampling rate $= \infty$ is from (Choi & Lee, 2000). For each resolution level we define the maximum control point absolute values $\gamma^{(k)}$ as $0.99 \times \frac{1}{K_n^{(k)}}$ (in our experiments we have $n = 3$ dimensional data). Codebase contains implementation for computing the value for other $k$.

| k | Sampling rate | $K_2^{(k)}$ | $K_3^{(k)}$ |
|---|---|---|---|
| 0 | 1 | 2.222222222 | 2.777777778 |
| 1 | 2 | 2.031168620 | 2.594390728 |
| 2 | 4 | 2.084187826 | 2.512366240 |
| 3 | 8 | 2.063570023 | 2.495476474 |
| 4 | 16 | 2.057074951 | 2.489089713 |
| 5 | 32 | 2.052177394 | 2.484247818 |
| 6 | 64 | 2.049330491 | 2.481890143 |
| 7 | 128 | 2.047871477 | 2.480726430 |
| 8 | 256 | 2.047136380 | 2.480102049 |
| $\infty$ | $\infty$ | 2.046392675 | 2.479472335 |

# B HYPERPARAMETER SELECTION DETAILS

We experimented on validation set with different hyperparameters during the development. While the final results on test set are computed only for one chosen configuration, the results on validation set might still be of interest for the reader. Results of these experiments for the OASIS dataset are shown in Table 5 and for the LPBA40 dataset in Table 6.

For the OASIS dataset we experimented with two configurations of number of resolution levels $K$. With both of these configurations we tested three different values for the regularization weight $\lambda$.

For the LPBA40 dataset we experimented with total of 6 configurations of number of resolution levels $K$ and whether to predict an affine transformation, but used the regularization weight value $\lambda = 1.0$ for all of them.

With the raw OASIS dataset without pre-alignment we used 6 resolution levels, together with an affine transformation prediction stage before the other deformation updates. We omitted the predicted affine transformation from the deformation regularization.

Table 5: Hyperparameter optimization results for our method calculated on the OASIS validation set. The chosen configuration was $\lambda = 1.0$, and $K = 4$. HD95 metric is not included due to relatively high computational cost.

| Hyperparameters | | Accuracy | Deformation regularity | | Consistency | |
|---|---|---|---|---|---|---|
| $\lambda$ | $K$ | Dice $\uparrow$ | $|J_\phi|_{\leq 0} \downarrow$ | $\text{std}(|J_\phi|) \downarrow$ | Cycle $\downarrow$ | Inverse $\downarrow$ |
| 1.0 | 5 | 0.822(0.035) | 9.1e−3(1.7e−3) | 0.45(0.027) | 5.7e−3(6.0e−4) | 5.7e−3(6.0e−4) |
| 1.5 | 5 | 0.818(0.034) | 1.9e−3(5.1e−4) | 0.40(0.023) | 3.7e−3(3.4e−4) | 3.7e−3(3.4e−4) |
| 2.0 | 5 | 0.815(0.035) | 3.7e−4(2.0e−4) | 0.37(0.021) | 2.6e−3(2.1e−4) | 2.6e−3(2.1e−4) |
| 1.0 | 4 | 0.822(0.034) | 8.2e−3(1.5e−3) | 0.44(0.028) | 5.5e−3(5.6e−4) | 5.5e−3(5.6e−4) |
| 1.5 | 4 | 0.819(0.035) | 2.1e−3(5.8e−4) | 0.40(0.023) | 3.4e−3(3.3e−4) | 3.4e−3(3.3e−4) |
| 2.0 | 4 | 0.815(0.036) | 3.6e−4(2.1e−4) | 0.37(0.020) | 2.6e−3(2.2e−4) | 2.6e−3(2.2e−4) |

Table 6: Hyperparameter optimization results for our method calculated on the LPBA40 validation set. The chosen configuration was $\lambda = 1.0$, $K = 7$, and Affine = No.

| Hyperparameters | | | Accuracy | | Deformation regularity | | Consistency | |
|---|---|---|---|---|---|---|---|---|
| $\lambda$ | $K$ | Affine | Dice $\uparrow$ | HD95 $\downarrow$ | $|J_\phi|_{\leq 0} \downarrow$ | $\text{std}(|J_\phi|) \downarrow$ | Cycle $\downarrow$ | Inverse $\downarrow$ |
| 1.0 | 4 | No | 0.710(0.015) | 6.10(0.46) | 2.5e−3(8.7e−4) | 0.31(0.020) | 2.5e−3(3.5e−4) | 2.5e−3(3.5e−4) |
| 1.0 | 5 | No | 0.720(0.014) | 5.83(0.36) | 1.7e−3(5.7e−4) | 0.30(0.019) | 2.3e−3(3.1e−4) | 2.3e−3(3.1e−4) |
| 1.0 | 6 | No | 0.725(0.012) | 5.70(0.31) | 2.1e−3(4.8e−4) | 0.29(0.019) | 2.3e−3(3.0e−4) | 2.3e−3(3.0e−4) |
| 1.0 | 7 | No | 0.726(0.011) | 5.69(0.30) | 1.9e−3(5.3e−4) | 0.29(0.019) | 2.3e−3(3.1e−4) | 2.3e−3(3.1e−4) |
| 1.0 | 5 | Yes | 0.719(0.014) | 5.86(0.35) | 2.2e−3(7.2e−4) | 0.30(0.019) | 2.4e−3(3.3e−4) | 2.4e−3(3.3e−4) |
| 1.0 | 6 | Yes | 0.721(0.015) | 5.78(0.37) | 2.6e−3(5.8e−4) | 0.30(0.019) | 2.4e−3(3.2e−4) | 2.4e−3(3.2e−4) |

## C  HYPERPARAMETER SELECTION DETAILS FOR BASELINES

For cLapIRN baseline we used the regularization parameter value $\overline{\lambda} = 0.05$ for the OASIS dataset and value $\overline{\lambda} = 0.1$ for the LPBA40 dataset where $\overline{\lambda}$ is used as in the paper presenting the method (Mok & Chung, 2021). The values were chosen based on the validation set results shown in Tables 7 and 8.

We trained VoxelMorph with losses and regularization weight identical to our method and for SYM-Net we used hyperparameters directly provided by Mok & Chung (2020a). We used the default number of convolution features for the baselines except for VoxelMorph we doubled the number of features, as that was suggested in the paper (Balakrishnan et al., 2019).

Table 7: Regularization parameter optimization results for cLapIRN calculated on the OASIS validation set. Here $\overline{\lambda}$ refers to the normalized regularization weight of the gradient loss of cLapIRN and should be in range $[0, 1]$. Value $\overline{\lambda} = 0.05$ was chosen since it resulted in clearly the highest Dice score. HD95 metric is not included due to relatively high computational cost.

| Hyperparameters | Accuracy | Deformation regularity | | Consistency |
|---|---|---|---|---|
| $\overline{\lambda}$ | Dice $\uparrow$ | $\|J_\phi\|_{\leq 0} \downarrow$ | $\mathrm{std}(\|J_\phi\|) \downarrow$ | Cycle $\downarrow$ |
| 0.01 | 0.812(0.034) | 2.5e0(2.9e−1) | 0.82(0.048) | 1.7e0(1.5e−1) |
| 0.05 | 0.817(0.034) | 1.1e0(1.8e−1) | 0.56(0.029) | 1.2e0(1.3e−1) |
| 0.1 | 0.812(0.035) | 4.2e−1(1.1e−1) | 0.43(0.020) | 8.9e−1(1.1e−1) |
| 0.2 | 0.798(0.038) | 7.2e−2(3.9e−2) | 0.30(0.013) | 6.0e−1(8.3e−2) |
| 0.4 | 0.769(0.042) | 1.4e−3(1.7e−3) | 0.18(0.0087) | 3.5e−1(4.4e−2) |
| 0.8 | 0.727(0.049) | 3.4e−6(2.2e−5) | 0.10(0.0050) | 2.5e−1(3.8e−2) |
| 1.0 | 0.711(0.052) | 1.3e−6(1.7e−5) | 0.082(0.0042) | 2.3e−1(3.8e−2) |

Table 8: Regularization parameter optimization results for cLapIRN calculated on the LPBA40 validation set. Here $\overline{\lambda}$ refers to the normalized regularization weight of the gradient loss of cLapIRN and should be in range $[0, 1]$. Value $\overline{\lambda} = 0.1$ was chosen due to the best overall performance.

| Hyperparameters | Accuracy | | Deformation regularity | | Consistency |
|---|---|---|---|---|---|
| $\overline{\lambda}$ | Dice $\uparrow$ | HD95 $\downarrow$ | $\|J_\phi\|_{\leq 0} \downarrow$ | $\mathrm{std}(\|J_\phi\|) \downarrow$ | Cycle $\downarrow$ |
| 0.01 | 0.714(0.014) | 9.9e−1(1.5e−1) | 0.45(0.029) | 0.45(0.029) | 9.9e−1(2.2e−1) |
| 0.05 | 0.715(0.014) | 3.2e−1(6.8e−2) | 0.33(0.018) | 0.33(0.018) | 8.0e−1(2.1e−1) |
| 0.1 | 0.714(0.014) | 7.4e−2(2.4e−2) | 0.25(0.012) | 0.25(0.012) | 6.6e−1(2.1e−1) |
| 0.2 | 0.709(0.015) | 4.4e−3(2.4e−3) | 0.19(0.0090) | 0.19(0.0090) | 4.9e−1(1.9e−1) |
| 0.4 | 0.698(0.017) | 3.5e−5(5.7e−5) | 0.13(0.0071) | 0.13(0.0071) | 3.6e−1(1.9e−1) |
| 0.8 | 0.678(0.019) | 5.0e−6(2.2e−5) | 0.085(0.0062) | 0.085(0.0062) | 3.0e−1(1.9e−1) |
| 1.0 | 0.671(0.021) | 5.0e−6(2.2e−5) | 0.074(0.0061) | 0.074(0.0061) | 3.0e−1(1.9e−1) |

## D  PROOF OF THEORETICAL PROPERTIES

While in the main text dependence of the intermediate outputs $d_{1\to1.5}^{(k)}$, $d_{2\to1.5}^{(k)}$, $z_1^{(k)}$, $z_2^{(k)}$, and $\delta^{(k)}$ on the input images $x_A, x_B$ is not explicitly written, throughout this proof we include the dependence in the notation since it is relevant for the proof.

### D.1  INVERSE CONSISTENT BY CONSTRUCT (THEOREM 3.1)

*Proof.* Inverse consistency by construct follows directly from Equation 8:

$$
\begin{aligned}
f_{1\to2}(x_A, x_B) &= d_{1\to1.5}^{(0)}(x_A, x_B) \circ d_{2\to1.5}^{(0)}(x_A, x_B)^{-1} \\
&= \left( d_{2\to1.5}^{(0)}(x_A, x_B) \circ d_{1\to1.5}^{(0)}(x_A, x_B)^{-1} \right)^{-1} \\
&= f_{2\to1}(x_A, x_B)^{-1}
\end{aligned}
$$

$\square$

Note that due to limited sampling resolution the inverse consistency error is not exactly zero despite of the proof. The same is true for earlier inverse consistent by construct registration methods, as discussed in Section 1.

To be more specific, sampling resolution puts a limit on the accuracy of the inverses obtained using deformation inversion layer, and also limits accuracy of compositions if deformations are resampled to their original resolution as part of the composition operation (see Section 3.6). While another possible source could be the fixed point iteration in deformation inversion layer converging imperfectly, that can be proven to be insignificant. As shown by Appendix E, the fixed point iteration is guaranteed to converge, and error caused by the lack of convergence of fixed point iteration can hence be controlled by the stopping criterion. In our experiments we used as a stopping criterion maximum inversion error within all the sampling locations reaching below one hundredth of a voxel, which is very small.

## D.2 Symmetric by construct (Theorem 3.2)

*Proof.* We use induction. Assume that for any $x_A$ and $x_B$ at level $k + 1$ the following holds: $d_{1\to1.5}^{(k+1)}(x_A, x_B) = d_{2\to1.5}^{(k+1)}(x_B, x_A)$. For level $K$ it holds trivially since $d_{1\to1.5}^{(K)}(x_A, x_B)$ and $d_{2\to1.5}^{(K)}(x_A, x_B)$ are defined as identity deformations. Using the induction assumption we have at level $k$:

$$z_1^{(k)}(x_A, x_B) = h^{(k)}(x_A) \circ d_{1\to1.5}^{(K)}(x_A, x_B) = h^{(k)}(x_A) \circ d_{2\to1.5}^{(K)}(x_B, x_A) = z_2^{(k)}(x_B, x_A)$$

Then also:

$$\begin{aligned}
\delta^{(k)}(x_A, x_B) &= u^{(k)}(z_1^{(k)}(x_A, x_B), z_2^{(k)}(x_A, x_B)) \circ u^{(k)}(z_2^{(k)}(x_A, x_B), z_1^{(k)}(x_A, x_B))^{-1} \\
&= u^{(k)}(z_2^{(k)}(x_B, x_A), z_1^{(k)}(x_B, x_A)) \circ u^{(k)}(z_1^{(k)}(x_B, x_A), z_2^{(k)}(x_B, x_A))^{-1} \\
&= \left[ u^{(k)}(z_1^{(k)}(x_B, x_A), z_2^{(k)}(x_B, x_A)) \circ u^{(k)}(z_2^{(k)}(x_B, x_A), z_1^{(k)}(x_B, x_A))^{-1} \right]^{-1} \\
&= \delta^{(k)}(x_B, x_A)^{-1}
\end{aligned}$$

Then we can finalize the induction step:

$$\begin{aligned}
d_{1\to1.5}^{(k)}(x_A, x_B) &= d_{1\to1.5}^{(k+1)}(x_A, x_B) \circ \delta^{(k)}(x_A, x_B) \\
&= d_{2\to1.5}^{(k+1)}(x_B, x_A) \circ \delta^{(k)}(x_B, x_A)^{-1} = d_{2\to1.5}^{(k)}(x_B, x_A)
\end{aligned}$$

From this follows that the method is symmetric by construct:

$$\begin{aligned}
f_{1\to2}(x_A, x_B) &= d_{1\to1.5}^{(0)}(x_A, x_B) \circ d_{2\to1.5}^{(0)}(x_A, x_B)^{-1} \\
&= d_{2\to1.5}^{(0)}(x_B, x_A) \circ d_{1\to1.5}^{(0)}(x_B, x_A)^{-1} = f_{2\to1}(x_B, x_A)
\end{aligned}$$

$\square$

The proven relation holds exactly.

## D.3 Topology preserving (Theorem 3.3)

*Proof.* As shown by Appendix E, each $u^{(k)}$ produces topology preserving (invertible) deformations (architecture of $u^{(k)}$ is described in Appendix A). Since the overall deformation is composition of multiple outputs of $u^{(k)}$ and their inverses, the whole deformation is then also invertible, and the method is topology preserving. $\square$

The inveritibility is not perfect if the compositions of $u^{(k)}$ and their inverses are resampled to the input image resolution, as is common practice in image registration. However, invertibility everywhere can be achieved by storing all the individual deformations and evaluating the composed deformation as their true composition (see Section 3.6 on inference variants and the results in Section 5).

# E  DERIVING THE OPTIMAL BOUND FOR CONTROL POINTS

As discussed in Section A, we limit absolute values of the predicted cubic spline control points defining the displacement field by a hard constraint $\gamma^{(k)}$ for each resolution level $k$. We want to find optimal $\gamma^{(k)}$ which ensure invertibility of individual deformations and convergence of the fixed point iteration in deformation inversion layer. We start by proving the continuous case and then extend it to our sampling based case. Note that the proof provides a significantly shorter and more general proof of the theorems 1 and 4 in (Choi & Lee, 2000).

By (Chen et al., 2008) a deformation field is invertible by the proposed deformation inversion layer (and hence invertible in general) if its displacement field is contractive mapping with respect to some norm (the convergence is then also with respect to that norm). In finite dimensions convergence in any p-norm is equal and hence we should choose the norm which gives the loosest bound.

Let us choose $||\cdot||_\infty$ norm for our analysis, which, as it turns out, gives the loosest possible bound. Then the Lipschitz constant of a displacement field is equivalent to the maximum $||\cdot||_\infty$ operator norm of the local Jacobian matrices of the displacement field.

Since for matrices $||\cdot||_\infty$ norm corresponds to maximum absolute row sum it is enough to consider one component of the displacement field.

Let $B : \mathbb{R} \to \mathbb{R}$ be a centered cardinal B-spline of some degree (actually any continuous almost everywhere differentiable function with finite support is fine) and let us consider an infinite grid of control points $\phi : \mathbb{Z}^n \to \mathbb{R}$ where $n$ is the dimensionality of the displacement field. For notational convinience, let us define a set $N := \{1, \ldots, n\}$.

Now let $f_\phi$ be the $n$-dimensional displacement field (or one component of it) defined by the control point grid:

$$f_\phi(x) = \sum_{\alpha \in \mathbb{Z}^n} \phi(\alpha) \prod_{i \in N} B(x_i - \alpha_i) \tag{11}$$

Note that since the function $B$ has finite support the first sum over $\mathbb{Z}^n$ can be defined as a finite sum for any $x$ and is hence well-defined. Also, without loss of generality it is enough to look at region $x \in [0, 1]^n$ due to the unit spacing of the control point grid.

For partial derivatives $\frac{\partial f_\phi}{\partial x_j} : \mathbb{R}^n \to \mathbb{R}$ we have

$$\frac{\partial f_\phi}{\partial x_j}(x) := \sum_{\alpha \in \mathbb{Z}^n} \phi(\alpha) \, B'(x_j - \alpha_j) \prod_{i \in N \setminus \{j\}} B(x_i - \alpha_i) = \sum_{\alpha \in \mathbb{Z}^n} \phi_\alpha D^j(x - \alpha) \tag{12}$$

where $D^j(x - \alpha) := B'(x_j - \alpha_j) \prod_{i \in N \setminus \{j\}} B(x_i - \alpha_i)$.

Following the power set notation, let us denote control points limited to some set $S \subset \mathbb{R}$ as $S^{\mathbb{Z}^n}$. That is, if $\phi \in S^{\mathbb{Z}^n}$, then for all $\alpha \in \mathbb{Z}^n$, $\phi(\alpha) \in S$.

**Lemma E.1.** *For all $\phi \in \,] -1/\tilde{K}_n, 1/\tilde{K}_n[^{\mathbb{Z}^n}$, $f_\phi$ is a contractive mapping with respect to the $||\cdot||_\infty$ norm, where*

$$\tilde{K}_n := \max_{\substack{x \in [0,1]^n \\ \tilde{\phi} \in [-1,1]^{\mathbb{Z}^n}}} \sum_{j \in N} \left| \frac{\partial f_{\tilde{\phi}}}{\partial x_j}(x) \right|. \tag{13}$$

*Proof.* For all $x \in [0,1]^n$, $\phi \in \, ]-1/\tilde{K}_n, 1/\tilde{K}_n[^{\mathbb{Z}^n}$

$$
\begin{aligned}
\sum_{j \in N} \left| \frac{\partial f_\phi}{\partial x_j}(x) \right| &< \max_{\substack{\tilde{x} \in [0,1]^n \\ \tilde{\phi} \in [-1/\tilde{K}_n, 1/\tilde{K}_n]^{\mathbb{Z}^n}}} \sum_{j \in N} \left| \frac{\partial f_{\tilde{\phi}}}{\partial \tilde{x}_j}(\tilde{x}) \right| \\
&= \max_{\substack{\tilde{x} \in [0,1]^n \\ \tilde{\phi} \in [-1,1]^{\mathbb{Z}^n}}} \sum_{j \in N} \left| \frac{\partial f_{\tilde{\phi}/\tilde{K}_n}}{\partial \tilde{x}_j}(\tilde{x}) \right| \\
&= \frac{1}{\tilde{K}_n} \max_{\substack{\tilde{x} \in [0,1]^n \\ \tilde{\phi} \in [-1,1]^{\mathbb{Z}^n}}} \sum_{j \in N} \left| \frac{\partial f_{\tilde{\phi}/\tilde{K}_n}}{\partial \tilde{x}_j}(\tilde{x}) \right| = \frac{\tilde{K}_n}{\tilde{K}_n} = 1.
\end{aligned}
\tag{14}
$$

Sums of absolute values of partial derivatives are exactly the $\|\cdot\|_\infty$ operator norms of the local Jacobian matrices of $f$, hence $f$ is a contraction. $\qquad\square$

**Lemma E.2.** *For any $k \in N$, $x \in [0,1]^n$, $\phi \in [-1,1]^{\mathbb{Z}^n}$, we can find some $\tilde{x} \in [0,1]^n$, $\tilde{\phi} \in [-1,1]^{\mathbb{Z}^n}$ such that*

$$
\frac{\partial f_{\tilde{\phi}}}{\partial x_j}(\tilde{x}) = \begin{cases} -\frac{\partial f_\phi}{\partial x_j}(x) & \text{for } j = k \\ \frac{\partial f_\phi}{\partial x_j}(x) & \text{for } j \in N \setminus \{k\}. \end{cases}
\tag{15}
$$

*Proof.* The B-splines are symmetric around origin:

$$
B(x) = B(-x) \implies B'(x) = -B'(-x)
\tag{16}
$$

Let us propose

$$
\tilde{x}_i := \begin{cases} 1 - x_i, & \text{when } i \in N \setminus k \\ x_i, & \text{when } i = k \end{cases}
\tag{17}
$$

and $\tilde{\phi} : \mathbb{Z}^n \to \mathbb{R}$ as $\tilde{\phi}(\alpha) := -\phi(g(\alpha))$ where $g : \mathbb{Z}^n \to \mathbb{Z}^n$ is a bijection defined as follows:

$$
g(\alpha)_i := \begin{cases} 1 - \alpha_i, & \text{when } i \in N \setminus k \\ \alpha_i, & \text{when } i = k. \end{cases}
\tag{18}
$$

Then for all $\alpha \in \mathbb{Z}^n$:

$$
\begin{aligned}
D^k(\tilde{x} - \alpha) &= B'(\tilde{x}_k - \alpha_k) \prod_{i \in N \setminus \{k\}} B(\tilde{x}_i - \alpha_i) \\
&= B'(x_k - g(\alpha)_k) \prod_{i \in N \setminus \{k\}} B(-(x_i - g(\alpha)_i)) \\
&= D^k(x - g(\alpha))
\end{aligned}
\tag{19}
$$

which gives

$$
\begin{aligned}
\frac{\partial f_{\tilde{\phi}}}{\partial \tilde{x}_k}(\tilde{x}) &= \sum_{\alpha \in \mathbb{Z}^n} \tilde{\phi}(\alpha) D^k(\tilde{x} - \alpha) \\
&= \sum_{\alpha \in \mathbb{Z}^n} -\phi(g(\alpha)) D^k(x - g(\alpha)) \qquad \text{g is bijective} \\
&= -\frac{\partial f_\phi}{\partial x_k}(x).
\end{aligned}
\tag{20}
$$

And for all $j \in N \setminus \{k\}, \alpha \in \mathbb{Z}^n$

$$
\begin{aligned}
D^j(\tilde{x} - \alpha) &= B'(\tilde{x}_j - \alpha_j) \prod_{i \in N \setminus \{j\}} B(\tilde{x}_i - \alpha_i) \\
&= B'(\tilde{x}_j - \alpha_j) \, B(\tilde{x}_k - \alpha_k) \prod_{i \in N \setminus \{j,k\}} B(\tilde{x}_i - \alpha_i) \\
&= B'(-(x_j - g(\alpha)_j)) \, B(x_k - g(\alpha)_k) \prod_{i \in N \setminus \{j,k\}} B(-(x_i - g(\alpha)_i)) \\
&= -B'(x_j - g(\alpha)_j) \, B(x_k - g(\alpha)_k) \prod_{i \in N \setminus \{j,k\}} B(x_i - g(\alpha)_i) \\
&= -B'(x_j - g(\alpha)_j) \prod_{i \in N \setminus \{j\}} B(x_i - g(\alpha)_i) \\
&= -D^k(x - g(\alpha))
\end{aligned}
\tag{21}
$$

which gives for all $j \in N \setminus \{k\}$

$$
\begin{aligned}
\frac{\partial f_{\tilde{\phi}}}{\partial \tilde{x}_j}(\tilde{x}) &= \sum_{\alpha \in \mathbb{Z}^n} \tilde{\phi}(\alpha) D^j(\tilde{x} - \alpha) \\
&= \sum_{\alpha \in \mathbb{Z}^n} -\phi(g(\alpha)) - D^j(x - g(\alpha)) \qquad \text{g is bijective} \\
&= \frac{\partial f_{\phi}}{\partial x_j}(x).
\end{aligned}
\tag{22}
$$

$\square$

**Theorem E.3.** *For all $\phi \in \,] -1/K_n, 1/K_n[^{\mathbb{Z}^n}$, $f_\phi$ is a contractive mapping with respect to the $|| \cdot ||_\infty$ norm, where*

$$
K_n := \max_{x \in [0,1]^n} \sum_{\alpha \in \mathbb{Z}^n} \left| \sum_{j \in N} D_\alpha^j(x) \right|.
\tag{23}
$$

*Proof.* Let us show that $K_n = \tilde{K}_n$.

$$
\begin{aligned}
\tilde{K}_n &= \max_{\substack{x \in [0,1]^n \\ \phi \in [-1,1]^{\mathbb{Z}^n}}} \sum_{j \in N} \left| \frac{\partial f_\phi}{\partial x_j}(x) \right| \\
&= \max_{\substack{x \in [0,1]^n \\ \phi \in [-1,1]^{\mathbb{Z}^n}}} \sum_{j \in N} \frac{\partial f_\phi}{\partial x_j}(x) \qquad \text{(Lemma E.2)} \\
&= \max_{\substack{x \in [0,1]^n \\ \phi \in [-1,1]^{\mathbb{Z}^n}}} \sum_{j \in N} \sum_{\alpha \in \mathbb{Z}^n} \phi_\alpha \, D^j(x - \alpha) \\
&= \max_{\substack{x \in [0,1]^n \\ \phi \in [-1,1]^{\mathbb{Z}^n}}} \sum_{\alpha \in \mathbb{Z}^n} \phi_\alpha \sum_{j \in N} D^j(x - \alpha) \\
&= \max_{x \in [0,1]^n} \sum_{\alpha \in \mathbb{Z}^n} \left| \sum_{j \in N} D_\alpha^j(x) \right| = K_n
\end{aligned}
\tag{24}
$$

The last step follows from the obvious fact that the sum is maximized when choosing each $\phi_\alpha$ to be either $1$ or $-1$ based on the sign of the inner sum $\sum_{j \in N} D^j(x - \alpha)$.

By Lemma E.1 $f$ is then a contractive mapping with respect to the $|| \cdot ||_\infty$ norm. $\square$

Theorem E.3 proves that if we limit the control point absolute values to be less than $1/K_n$, then the resulting deformation is invertible by the fixed point iteration. Also, approximating $K_n$ accurately is possible at least for $n \leq 3$. Subset of $\mathbb{Z}^n$ over which the sum needs to be taken depends on the support of the function $B$ which again depends on the degree of the B-splines used.

Next we want to show that the obtained bound is also tight bound for invertibility of the deformation. That also then shows that $|| \cdot ||_\infty$ norm gives the loosest possible bound.

Since $f_\phi$ corresponds only to one component of a displacement field, let us consider a fully defined displacement field formed by stacking $n$ number of $f_\phi$ together. Let us define

$$g_\phi(x) := (f_\phi)_{i \in N}. \tag{25}$$

**Theorem E.4.** *There exists $\phi \in [-1/K_n, 1/K_n]^{\mathbb{Z}^n}$, $x \in [0,1]^n$ s.t. $\det\left(\frac{\partial g_\phi}{\partial x} + I\right)(x) = 0$ where $\frac{\partial g_\phi}{\partial x}$ is the Jacobian matrix of $g_\phi$ and $I$ is the identity matrix.*

*Proof.* By Lemma E.2 and Theorem E.3 there exists $x \in [0,1]^n$ and $\tilde{\phi} \in [-1,1]^{\mathbb{Z}^n}$ such that

$$\sum_{j \in N} \frac{\partial f_{\tilde{\phi}}}{\partial x_j}(x) = -K_n \tag{26}$$

where all $\frac{\partial f_{\tilde{\phi}}}{\partial x_j}(x) < 0$.

Let us define $\phi := \tilde{\phi}/K_n \in [-1/K_n, 1/K_n]^{\mathbb{Z}^n}$. Then

$$\sum_{j \in N} \frac{\partial f_\phi}{\partial x_j}(x) = -1. \tag{27}$$

Now let $y \in \mathbb{R}^n$ be a vector consisting only of values 1, that is $y =: (1)_{i \in N}$. Then one has

$$\begin{aligned}
\left(\frac{\partial g_\phi}{\partial x} + I\right)(x)y &= \left(\frac{\partial g_\phi}{\partial x}(x)\right) y + y \\
&= \left(\sum_{j \in N} 1 \frac{\partial f_\phi}{\partial x_j}(x)\right)_{i \in N} + (1)_{i \in N} \\
&= (-1)_{i \in N} + (1)_{i \in N} = 0.
\end{aligned} \tag{28}$$

In other words $y$ is an eigenvector of $\left(\frac{\partial g_\phi}{\partial x} + I\right)(x)$ with eigenvalue 0 meaning that the determinant of $\left(\frac{\partial g_\phi}{\partial x} + I\right)(x)$ is also 0. $\qquad \square$

The proposed bound is hence the loosest possible since the deformation can have zero Jacobian at the bound, meaning it is not invertible.

### E.1 SAMPLING BASED CASE (EQUATION 10)

The bound used in practice, given in Equation 10, is slightly different to the bound proven in Theorem E.3. The reason is that for computational efficiency we do not use directly the cubic B-spline representation for the displacement field but instead take only samples of the displacement field in the full image resolution (see Appendix A), and use efficient bi- or trilinear interpolation for defining the intermediate values. As a result the continuous case bound does not apply anymore.

However, finding the exact bounds for our approximation equals evaluating the maximum in Theorem E.3 over a finite set of sampling locations and replacing $D^j(x)$ with finite difference derivatives. The mathematical argument for that goes almost identically and will not be repeated here. However, to justify using finite difference derivatives, we need the following two trivial remarks:

- When defining a displacement field using grid of values and bi- or trilinear interpolation, the highest value for $|| \cdot ||_\infty$ operator norm is obtained at the corners of each interpolation patch.

- Due to symmetry, it is enough to check derivative only at one of $2^n$ corners of each bi- or trilinear interpolation patch in computing the maximum (corresponding to finite difference derivative in only one direction over each dimension).

Maximum is evaluated over the relative sampling locations with respect to the resolution of the control point grid (which is in the resolution of the features $z_1^{(k)}$ and $z_2^{(k)}$). The exact sampling grid depends on how the sampling is implemented (which is an implementation detail), and in our case we used the locations $X := \{1/2 + \frac{1}{2^{k+1}} + \frac{i}{2^k} \mid i \in \mathbb{Z}\}^n \cap [0,1]^n$ which have, without loss of generality, been again limited to the unit cube.

No additional insights are required to show that the equation 10 gives the optimal bound.

## F  DEFORMATION INVERSION LAYER MEMORY USAGE

We conducted an experiment on the memory usage of the deformation inversion layer compared to the stationary velocity field (SVF) framework (Arsigny et al., 2006) since SVF framework could also be used to implement the suggested architecture in practice.

With the SVF framework one could slightly simplify the deformation update Equation 5 to the form

$$U^{(k)} := \exp(u^{(k)}(z_1^{(k)}, z_2^{(k)}) - u^{(k)}(z_2^{(k)}, z_1^{(k)})) \tag{29}$$

where $\exp$ is the SVF integration (corresponding to Lie algebra exponentiation), and $u^{(k)}$ now predicts an auxiliary velocity field. We compared memory usage of this to our implementation, and used the implementation by Dalca et al. (2018) for SVF integration.

The results are shown in Table 9. Our version implemented using the deformation inversion layer requires 5 times less data to be stored in memory for the backward pass compared to the SVF integration. The peak memory usage during the inversion is also slightly lower. The memory saving is due to the memory efficient back-propagation through the fixed point iteration layers, which requires only the final inverted volume to be stored for backward pass. Since our architecture requires two such operations for each resolution level ($U^{(k)}$ and its inverse), the memory saved during training is significant.

Table 9: **Memory usage comparison between deformation inversion layer and stationary velocity field (SVF) based implementations.** The comparison is between executing Equation 5 using deformation inversion layers and executing Equation 29 using SVF integration implementation by Dalca et al. (2018). Between passes memory usage refers to the amount memory needed for storing values between forward and backward passes, and peak memory usage refers to the peak amount of memory needed during forward and backward passes. A volume of shape $(256, 256, 256)$ with 32 bit precision was used. We used 7 scalings and squarings for the SVF integration.

| Method | Between passes memory usage (GB) ↓ | Peak memory usage (GB) ↓ |
|---|---|---|
| Deformation inversion layer | **0.5625** | **3.9375** |
| SVF integration | 2.8125 | 4.125 |

## G  RELATED WORK

In this appendix we aim to provide a more thorough analysis of the related work, by introducing in more detail the works that we find closest to our method, and explaining how our method differs from those.

### G.1  CLASSICAL REGISTRATION METHODS

Classical registration methods, as opposed to the learning based methods, optimize the deformation independently for any given single image pair. For this reason they are sometimes called optimization based methods.

**DARTEL** by Ashburner (2007) is a classical optimization based registration method built on top of the stationary velocity field (SVF) (Arsigny et al., 2006) framework offering symmetric by construct, inverse consistent, and topology preserving registration. The paper is to our knowledge the first symmetric by construct, inverse consistent, and topology preserving registration method.

**SyN** by Avants et al. (2008) is another classical symmetric by construct, inverse consistent, and topology preserving registration method. The properties are achieved by the Large Deformation Diffeomorphic Metric Mapping LDDMM framework (Beg et al., 2005) in which differoemphisms are generated from time-varying velocity fields (as opposed to the stationary ones in the SVF framework). The LDDMM framework has not been used much in unsupervised deep learning for generating diffeomorphims due to its computational cost. While some works exist (Wang & Zhang, 2020; Wu et al., 2022; Joshi & Hong, 2023) they have had to make significant modifications to the LDDMM framework, and as a result they have lost the by construct topology preserving properties. SyN is to our knowledge the first work suggesting matching the images in the intermediate coordinates for achieving symmetry, the idea which was also used in our work. However, the usual implementation of SyN in ANTs (Avants et al., 2009) is not as a whole symmetric since the affine registration is not applied in a symmetric manner.

SyN has performed well in evaluation studies between different classical registration methods. e.g. (Klein et al., 2009). However, it is significantly slower than the strong baselines included in our study, and has already earlier been compared with those(Balakrishnan et al., 2019; Mok & Chung, 2020a;b), and hence was not included in our study.

### G.2  DEEP LEARNING METHODS (EARLIER WORK)

Unlike the optimization based methods above, deep learning methods train a neural network that, for two given input images, outputs a deformation directly. The benefits of this class of methods include the significant speed improvement and more robust performance (avoiding local optima) (De Vos et al., 2019). Our model belongs to this class of methods.

**SYMNet** by Mok & Chung (2020a) uses as single forward pass of a U-Net style neural network to predict two stationary velocity fields, $v_{1\to1.5}$ and $v_{2\to1.5}$ (in practice two 3 channeled outputs are extracted from the last layer features using separate convolutions). The stationary velocity fields are integrated into two half-way deformations (and their inverses). The training loss matches the images both in the intermediate coordinates and in the original coordinate spaces (using the composed full deformations). While the network is a priori symmetric with respect to the input images, changing the input order of of the images (concatenating the inputs in the opposite order for the U-Net) can in principle result in any two $v_{1\to1.5}$ and $v_{2\to1.5}$ (instead of swapping them), meaning that the method is not symmetric by construct as defined in Section 1 (this is confirmed by the cycle consistency experiments).

Our method does not use the stationary velocity field (SVF) framework to invert the deformations, but instead uses the novel deformation inversion layer. Also, SYMNet does not employ the multiresolution strategy. The use of intermediate coordinates is similar to our work.

**MICS** by Estienne et al. (2021) uses a shared convolutional encoder $E$ to encode both input images into some feature representations $E(x_A)$ and $E(x_B)$. A convolutional decoder network $D$ is then used to extract gradient volumes (constrained to contain only positive values) of the deformations for both forward and inverse deformations with formulas

$$\nabla f(x_A, x_B)_{1\to2} = D(E(x_A) - E(x_B)) \text{ and } \nabla f(x_A, x_B)_{2\to1} = D(E(x_B) - E(x_A)). \quad (30)$$

The final deformations are obtained from the gradient volumes by a cumulative sum operation. Since gradients are constrained to be positive the resulting deformation will be roughly invertible. However, as stated in their work, this only puts a positive constraint on the diagonal of the Jacobians, not on its determinant, unlike our work which guarantees positive determinants (Theorem 3.3).

While MICS is symmetric by construct in the sense that swapping $x_A$ and $x_B$ will result in swapping the predicted forward and inverse deformations, this symmetry is achieved by subtraction (Equation 30) instead of mathematical inverse operation (as in our work, Equation 2). As a result the predicted "forward" and "inverse" deformations are not actually by construct constrained to be forward and inverse deformations of each other. MICS uses a loss to enforce this. Also, while MICS employs a multi-step approach, the symmetric by construct property is lost over the whole architecture due to not using the intermediate coordinate approach employed by our work.

**Justification for baselines included in our work**: In addition to SYMNet, **cLapIRN**(Mok & Chung, 2021) was chosen as a baseline because it was the best method on OASIS dataset in the Learn2Reg challenge (Hering et al., 2022). It employs a standard and straightforward multi-resolution approach, and is not topology preserving, inverse consistent, or symmetric. Apart from the multi-resolution approach, it is not methodologically close to our method. **VoxelMorph** (Balakrishnan et al., 2019) is a standard baseline in deep learning based unsupervised registration, and it is based on a straightforward application of U-Net architecture to image registration.

### G.3 DEEP LEARNING METHODS (RECENT METHODS PARALLEL WITH OUR WORK)

These methods are very recent deep learning based registrations methods which have been developed independently of and in parallel with our work. A comparison with these methods and our model is therefore a fruitful topic for future research.

**Iglesias (2023)** uses a similar approach to achieve symmetry as our work but in the stationary velocity field (SVF) framework. In SVF framework, given some velocity field $v$, we have the property that $\exp(v) = \exp(-v)^{-1}$ where $\exp$ represents the integration operation (Arsigny et al., 2006) generating the deformation from the velocity field. Hence to obtain symmetric by construct method, one can modify the formula 2 to the form

$$f(x_A, x_B) := \exp(u(x_A, x_B) - u(x_B, x_A)). \quad (31)$$

which will result in a method which is symmetric by construct, inverse consistent, and topology preserving. We measure memory usage against this formulation in Appendix F and show that our formulation using the novel deformation inversion layer requires storing 5 times less memory for the backward pass. Their method includes only a single registration step, and not the robust multi-resolution architecture like ours.

**Greer et al. (2023)** extends the approach the approach by Iglesias (2023) to multi-step formulation in very similar way to how we construct our multi-resolution architecture by using the intermediate coordinates (square roots of deformations). They also employ the SVF framework, as opposed to our work which uses the deformation inversion layers, which, as shown in Appendix F, requires storing 5 times less memory for the backward pass, which is significant for being able to train the multi-step network on large images (such as our OASIS raw data). Also, their paper treats the separate steps of multi-step architecture as independent whereas we develop very efficient multi-resolution formulation based on first extracting the multi-resolution features using ResNet-style encoder.

### G.4 NOVEL ASPECTS OF OUR METHOD

Finally, here we further elaborate and summarize the differences between our work and previous methods, and highlight the key novel aspects of our method.

Multiple works in the past have tried to enforce one or all of the properties (symmetricity, inverse consistency, topology preservation), either by a loss function or by construct. However, only the classical iterative algorithms have enforced all of them by construct (Ashburner, 2007; Avants et al.,

2008). These papers are very highly cited ( 8000 and  5000 citations), and their main objective parallels ours - to propose a method which fulfills the properties by construct. The challenge we set out to tackle in our work was to develop the first such method in the deep learning setting.

To summarize, our novel methodological innovations, which set our method apart from the methods discussed above, were the novel trick to make the method symmetric (Section 3.1) and the new implicit deformation inversion layer (with potential impact outside this subfield), which uses 5 times less memory than implementing the same idea in the SVF framework makes this possible. Additionally, we showed how to integrate these innovations with state-of-the-art architectural building blocks, especially the multi-resolution architecture, for a method whose overall performance exceeded that of the existing methods. To clarify, our main innovations could be integrated with simpler networks to ensure the desirable properties, but for the state-of-the-art performance the best available building blocks, including the multi-resolution architecture, need to be adopted.

Regarding the general idea of the multi-resolution architecture, we acknowledge that it is standard and not a contribution specific to our method. However, the main challenge with developing the symmetric, topology preserving, and inverse consistent multi-resolution approach is making this computationally feasible for the very large volumetric data in image registration. Our novel highly memory efficient implicit deformation inversion makes this possible. Ensuring that the deformation prediction networks output deformations that are invertible by the deformation inversion layer also required non-trivial mathematical work of obtaining the optimal invertibility bounds (Appendix E). Another partially novel aspect of our multi-resolution approach is the idea of first extracting features independently in multiple resolutions using a ResNet encoder and then using those features to gradually update the deformation. While Young et al. (2022) proposed a similar overall architecture (in an otherwise different supervised setting), in our architecture the information flows back from the lower to the higher resolution levels only through the deformations, as opposed to their heavier U-net style approach.

## H    Deformation inversion layer practical convergence

We conducted an experiment on the fixed point iteration convergence in the deformation inversion layers with the model trained on OASIS dataset. The results can be seen in Figure 4. The main result was that in the whole OASIS test set of 9591 pairs not a single deformation required more than 8 iterations for convergence. Deformations requiring 8 iterations were only $0.05\%$ of all the deformations and a significant majority of the deformations ($96\%$) required 2 to 5 iterations. In all the experiments, including this one, the stopping criterion for the iterations was maximum displacement error within the whole volume reaching below one hundredth of a voxel, which is a very small error.

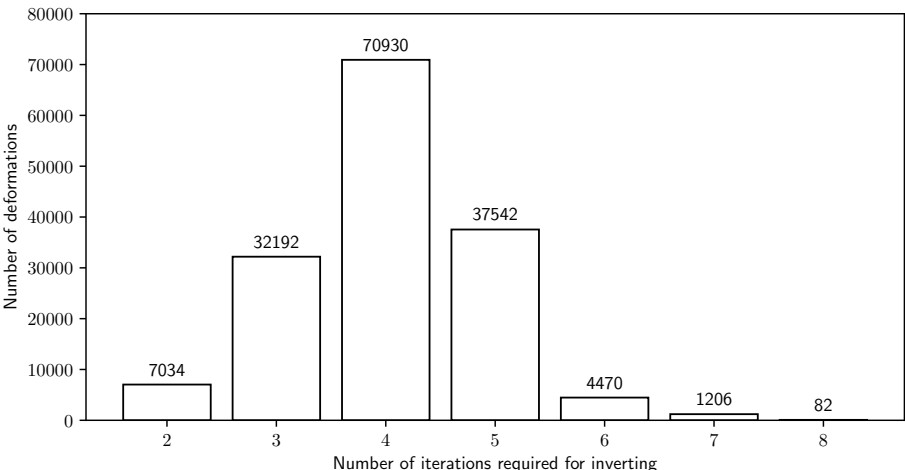

Figure 4: **Number of fixed point iterations required for convergence in deformation inversion layers with the model trained on OASIS dataset.** The stopping criterion for the fixed point iteration was maximum displacement error within the whole volume reaching below one hundredth of a voxel. All deformation inversions for the whole OASIS test set are included.

# I VISUALIZATIONS OF THE RESULTS

Figures 5, 6, and 7 visualize the differences in deformation regularity, cycle consistency, and inverse consistency respectively.

Figures 8 and 9 visualize dice scores for individual anatomical regions for both OASIS and LPBA40 datasets. VoxelMorph and SYMNet perform systematically worse than our method, while cLapIRN and our method perform very similarly on most regions.

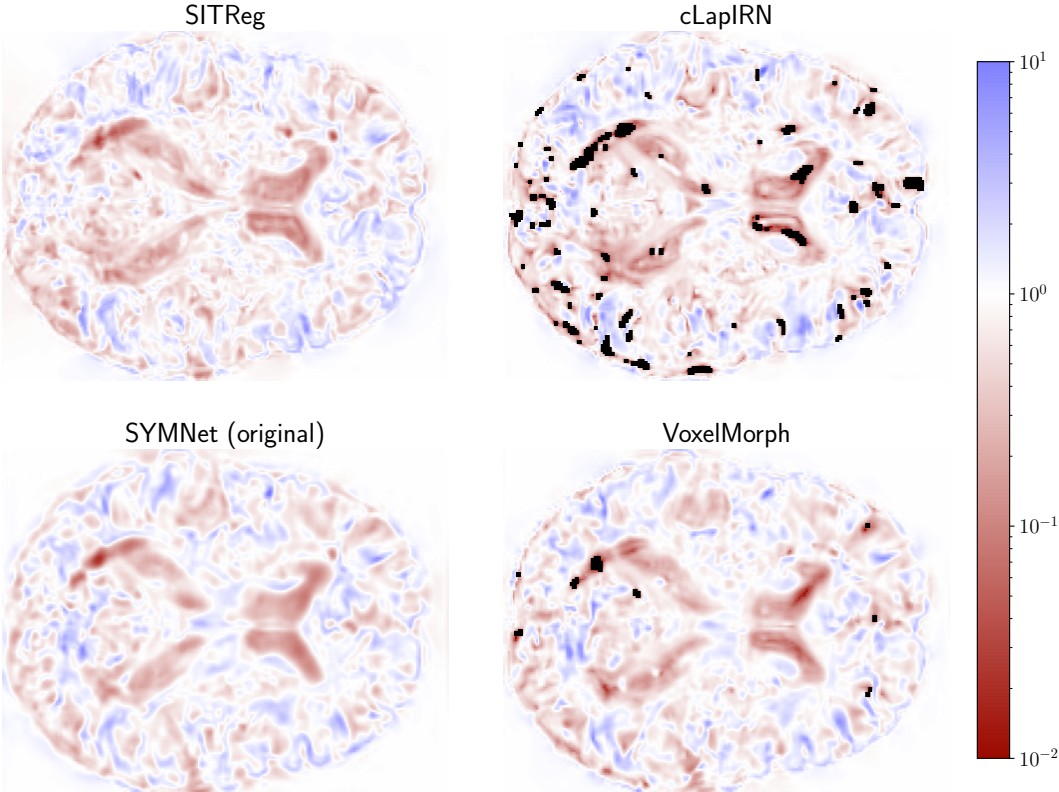

Figure 5: **Visual deformation regularity comparison.** Local Jacobian determinants are visualized for each model for a single predicted deformation in OASIS experiment. Folding voxels (determinant below zero) are marked with black color. Only one axial slice of the predicted 3D deformation is visible.

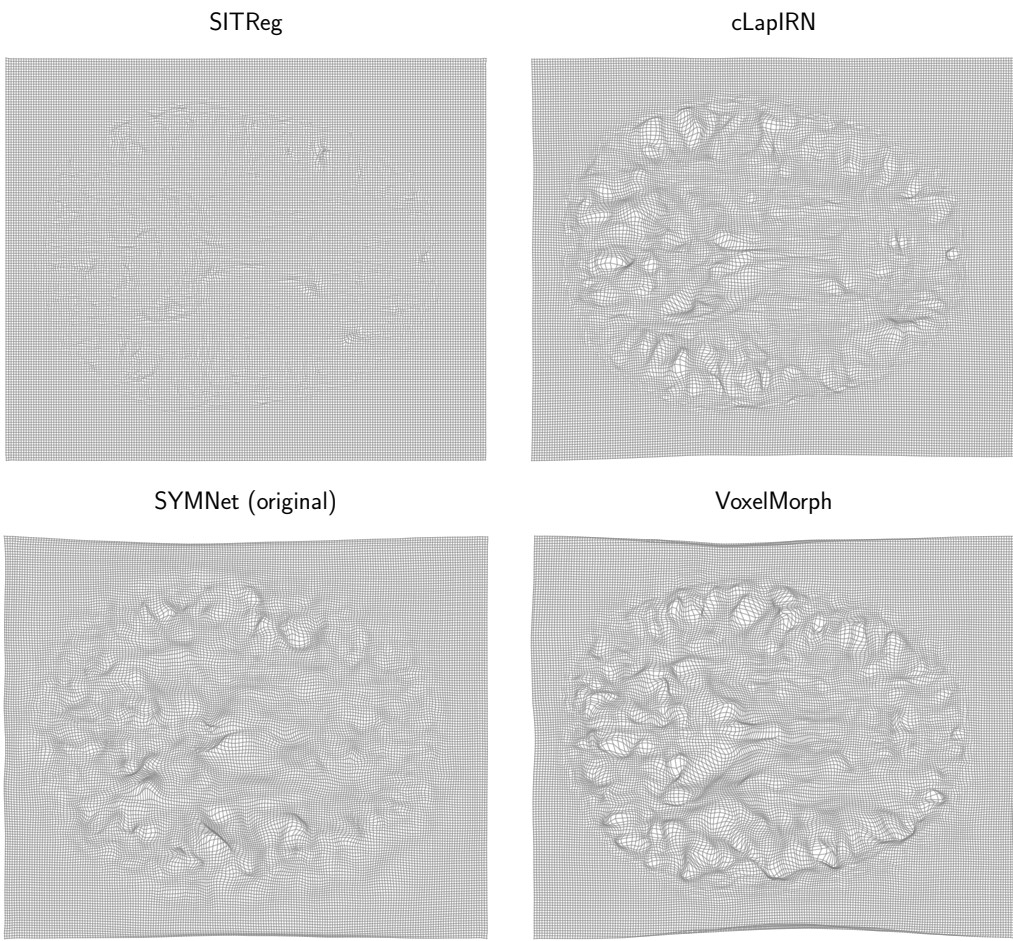

Figure 6: **Visual cycle consistency comparison.** The deformation composition $f(x_A, x_B) \circ f(x_B, x_A)$ is visualized for each model for a single image pair in LPBA40 experiment. Ideally, changing the order of the input images should result in the same coordinate mapping but in the inverse direction, since anatomical correspondence is not dependent on the input order. In other words, the deformation composition $f(x_A, x_B) \circ f(x_B, x_A)$ should equal the identity deformation. As can be seen, the property is only fulfilled (up to small sampling errors) by our method. Only one axial slice of the predicted 3D deformation is shown.

SITReg          SYMNet (original)

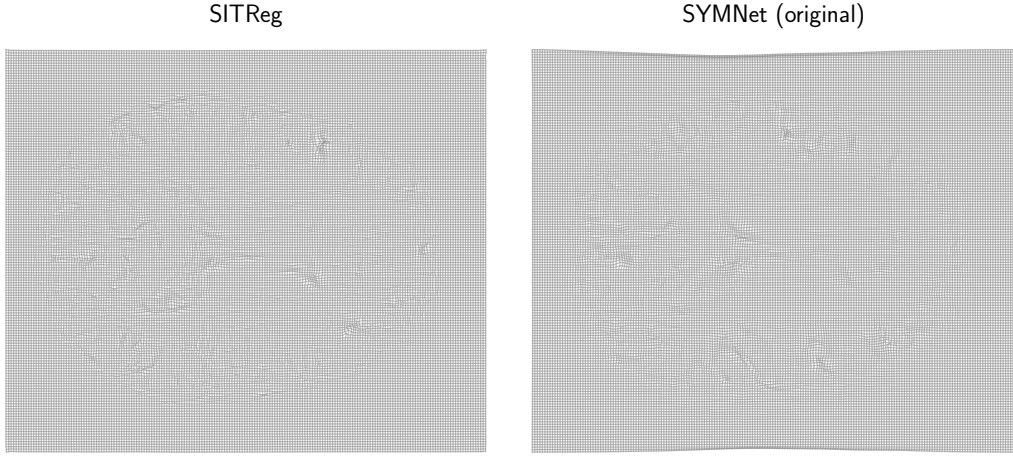

Figure 7: **Visual inverse consistency comparison.** The deformation $f(x_A, x_B)_{1\to 2} \circ f(x_A, x_B)_{2\to 1}$ is visualized for SITReg and SYMNet models for a single image pair in LPBA40 experiment. Since cLapIRN and VoxelMorph do not generate explicit inverses, they are not included in the figure. Ideally, $f(x_A, x_B)_{1\to 2} \circ f(x_A, x_B)_{2\to 1}$ should equal the identity mapping, and as can be seen, the property is well fulfilled by both of the methods. Only one axial slice of the predicted 3D deformation is visible.

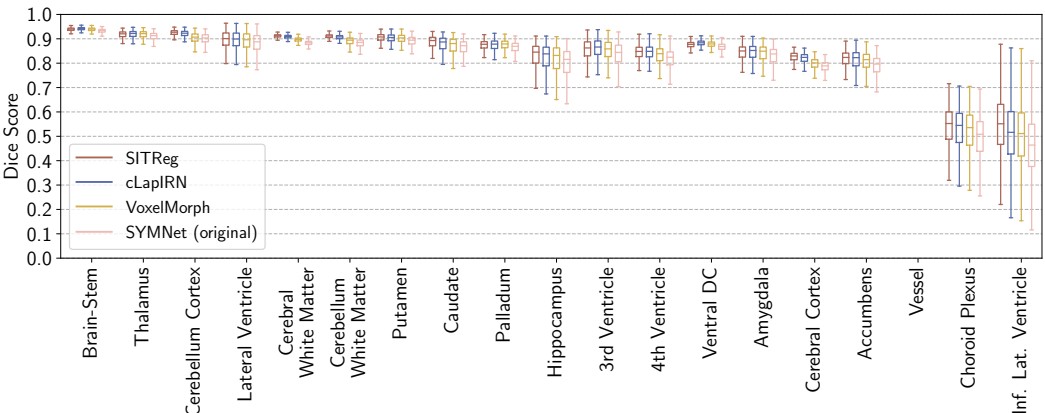

Figure 8: **Individual brain structure dice scores for the OASIS experiment.** Boxplot shows performance of each of the compared methods on each of the brain structures in the OASIS dataset. Algorithms from left to right in each group: SITReg, cLapIRN, VoxelMorph, SYMNet (original)

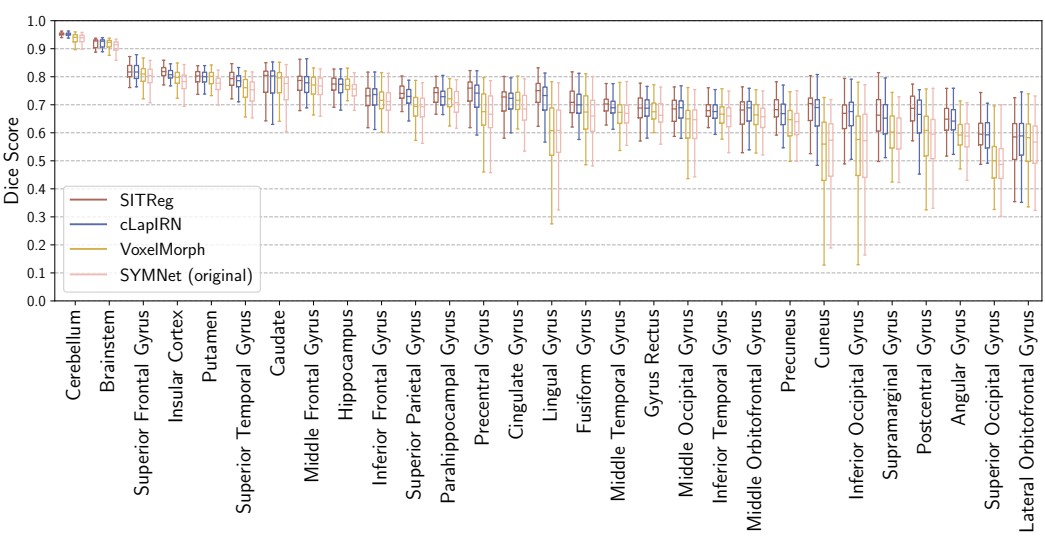

Figure 9: **Individual brain structure dice scores for the LPBA40 experiment.** Boxplot shows performance of each of the compared methods on each of the brain structures in the LPBA40 dataset. Algorithms from left to right in each group: SITReg, cLapIRN, VoxelMorph, SYMNet (original)

## J   ADDITIONAL VISUALIZATIONS

Figure 10 visualizes how the deformation is being gradually updated during the multi-resolution architecture.

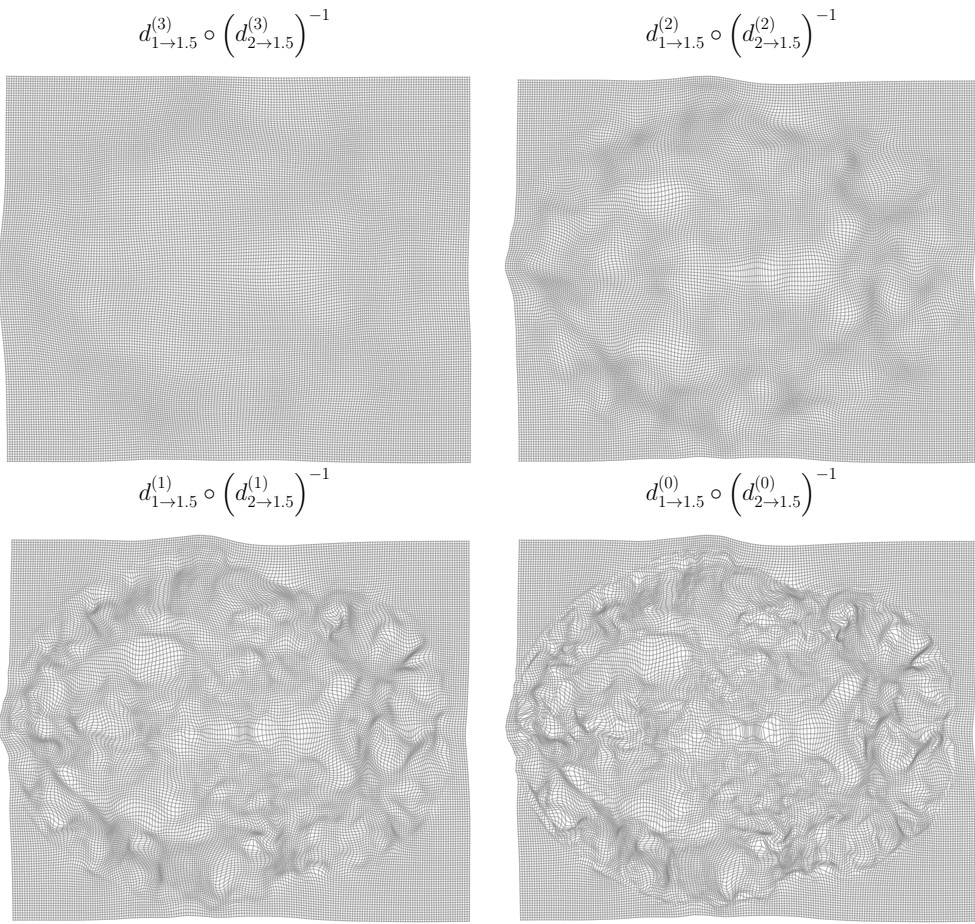

Figure 10: **Visualization of deformation being gradually updated.** Each $d_{1\to1.5}^{(k)} \circ \left(d_{2\to1.5}^{(k)}\right)^{-1}$ corresponds to the full deformation after resolution level $k$. The example is from the OASIS experiment.

## K   DATASET DETAILS

We split the OASIS dataset into 255, 20 and 139 images for training, validation, and testing. The split differs from the Learn2Reg challenge since the test set is not available, but sizes correspond to the splits used by Mok & Chung (2020a;b; 2021). We used all image pairs for testing and validation, yielding 9591 test and 190 validation pairs. For the affinely-aligned OASIS experiment we cropped the images to $144 \times 192 \times 160$ resolution. Images in raw OASIS dataset have resolution $256 \times 256 \times 256$ and we did not crop the images.

We split the LPBA40 into 25, 5 and 10 images for training, validation, and testing. This leaves us with 10 pairs for validation and 45 for testing. We cropped the LPBA40 images to $160 \times 192 \times 160$ resolution.

## L  DETAILS ON STATISTICAL SIGNIFICANCE

We computed statistical significance of the results comparing the test set predictions of the trained models with each other. We measured the statistical significance using permutation test, and in practice sampled 10000 permutations. In Figures 1 and 2 all the improvements denoted with asterisk (*) obtained very small p-value with not a single permutation (out of the 10000) giving larger mean difference than the one observed.

To establish for certain the relative performance of the methods with respect to the tight metrics, one should train multiple models per method with different random seeds. However, our claim is not that the developed method improves the results with respect to a single tight metric but rather that the overall performance is better by a clear margin (see Section 5).

## M  CLARIFICATIONS ON SYMMETRY, INVERSE CONSISTENCY, AND TOPOLOGY PRESERVATION

Here we provide examples of symmetry, inverse consistency and lack of topology preservation to further clarify how the terms are used in the paper.

Since symmetry and inverse consistency are quite similar properties, their exact difference might remain unclear. Examples of registration methods that are *inverse consistent by construct but not symmetric* are many deep learning frameworks applying the stationary velocity field (Arsigny et al., 2006) approach, e.g, (Dalca et al., 2018; Krebs et al., 2018; 2019; Mok & Chung, 2020a). All of them use a neural network to predict a velocity field for an ordered pair of input images. The final deformation is then produced via Lie algebra exponentiation of the velocity field, that is, by integrating the velocity field over itself over unit time. Details of the exponentiation are not important here but the operation has an interesting property: By negating the velocity field to be exponentiated, the exponentiation results in inverse deformation. Denoting the Lie algebra exponential by $\exp$, and using notation from Section 1, we can define such methods as

$$\begin{cases} f_{1 \to 2}(x_A, x_B) & := \exp(g(x_A, x_B)) \\ f_{2 \to 1}(x_A, x_B) & := \exp(-g(x_A, x_B)) \end{cases} \tag{32}$$

where $g$ is the learned neural network predicting the velocity field. As a result, the methods are inverse consistent by construct since $\exp(g(x_A, x_B)) = \exp(-g(x_A, x_B))^{-1}$ (accuracy is limited by spatial sampling resolution). However, by changing the order of inputs to $g$, there is no guarantee that $g(x_A, x_B) = -g(x_B, x_A)$ and hence such methods are not symmetric by construct.

MICS (Estienne et al., 2021) is an example of a method which is *symmetric by construct but not inverse consistent*. MICS is composed of two components: encoder, say $E$, and decoder, say $D$, both of which are learned. The method can be defined as

$$\begin{cases} f_{1 \to 2}(x_A, x_B) & := D(E(x_A, x_B) - E(x_B, x_A)) \\ f_{2 \to 1}(x_A, x_B) & := D(E(x_B, x_A) - E(x_A, x_B)). \end{cases} \tag{33}$$

As a result, the method is symmetric by construct since $f_{1 \to 2}(x_A, x_B) = f_{2 \to 1}(x_B, x_A)$ holds exactly. However, there is no architectural guarantee that $f_{1 \to 2}(x_A, x_B)$ and $f_{2 \to 1}(x_B, x_A)$ are inverses of each other, and the paper proposes to encourage that using a loss function. In the paper they use such components in multi-steps manner, and as a result the overall architecture is no longer symmetric.

Lack of topology preservation means in practice that the predicted deformation folds on top of itself. An example of such deformation is shown in Figure 11.

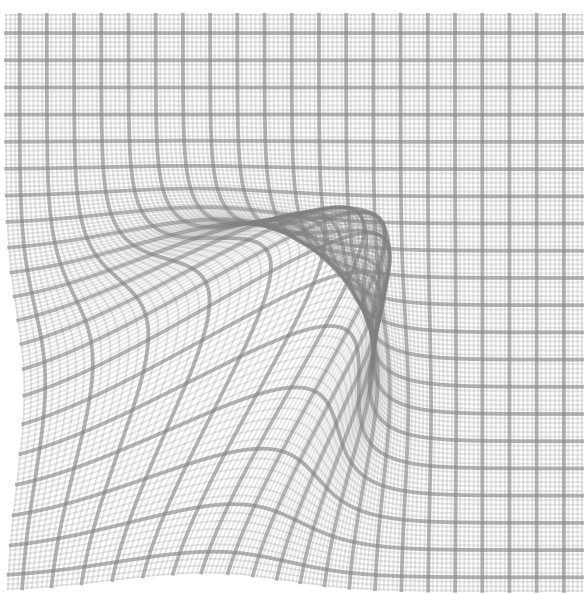

Figure 11: **Visualization of a 2D deformation which is not topology preserving.** The deformation can be seen folding on top of itself.

