# OpenReview forum: "SITReg: Multi-resolution architecture for symmetric, inverse consistent, and topology preserving image registration using deformation inversion layers"
_ICLR.cc/2024/Conference — Submitted to ICLR 2024_

### Official Review · Reviewer_eYpy · 2023-10-31

**Soundness:** 3 good
**Presentation:** 2 fair
**Contribution:** 2 fair
**Rating:** 6
**Confidence:** 4

**Summary:**

The paper proposes a deep learning-based approach to deformable image registration, which enforces symmetry, inverse consistency, and topology preservation. In contrast to previous approaches, which enforce these constraints via loss functions, the approach proposed here achieves this via construction. Additionally, the paper uses a multi-resolution feature representation for image registration. The approach is evaluated on the tasks of inter-subject brain MR registration, evaluated on the LPBA40 and Oasis datasets.

**Strengths:**

The paper addresses an important topic in biomedical image analysis. Developing robust and reliable methods for image registration is still an unsolved problem. The paper contains a good summary of the state-of-the-art in terms of previous publications. Similarly, the proposed method is compared against a number of strong baselines, including VoxelMorph, SYMNet and cLapIRN.

**Weaknesses:**

The symmetric formulation proposed in section 3.1 seems not entirely new. Indeed, the authors acknowledge this by stating that a similar approach has been used in recent registration methods (Estienne et al., 2021; Young et al., 2022). The multi-resolution formulation proposed in section 3.2 seems rather natural (also in the symmetric setting), so I am unclear on how this is different from standard multi-resolution formulations, which are ubiquitous in image registration settings.

The evaluation of the registration accuracy is limited to the assessment of Dice overlap and Hausdorff distance after registration. It would have been good if the authors had used some additional non-brain datasets which have landmarks (e.g. lung CT images from the EMPIRE10 challenge) and thus allow the calculation of quantities such as the target registration error. Furthermore, the improvements in registration accuracy seem rather than marginal. No visual examples of the registrations are provided.

**Questions:**

- What is the key novelty in section 3.2? How is this different from the traditional multi-resolution (except the symmetric formulation)?
- Can you comment on the significance of the improvement of the registration results?
- The advantage of symmetry and inverse consistency over loss-based approaches is not clear, as all methods seem to provide symmetric and inverse consistent registrations apart from numerical accuracy. Can you comment on this?

---

> ### Author Response · Authors · 2023-11-15
>
> **The symmetric formulation proposed in section 3.1 seems not entirely new. Indeed, the authors acknowledge this by stating that a similar approach has been used in recent registration methods (Estienne et al., 2021; Young et al., 2022).**
>
> It is true that Estienne et al. (2021) propose a by construct symmetric network. However,  their formulation is clearly different from the ideas in Section 3.1, and does not guarantee inverse or cycle consistency by construct. The method by Young et al. (2022) does not employ any approach close to our symmetric formulation, and we rather referred to it due to its similar multi-resolution feature extraction approach.
>
> **What is the key novelty in section 3.2? How is this different from the traditional multi-resolution (except the symmetric formulation)?**
>
> We agree that the principle itself behind the multi-resolution approach is rather standard and we have not claimed otherwise in our article.
>
> The main challenge with the multi-resolution approach was to make it computationally feasible for large volumetric data (as in image registration). For that, we developed the new implicit deformation inversion layer (which can have impact outside this subfield) which uses 5 times less memory than implementing the same idea in the SVF framework. Also, designing the deformation prediction networks $u^{(k)}$ to output deformations invertible by the deformation inversion layer required non-trivial mathematical work of obtaining the optimal invertibility bounds (Appendix E).
>
> The idea of first extracting features independently in multiple resolutions using a ResNet encoder and then using those features to gradually update the deformation is also partially new. While Young et al. (2022) proposed a similar architecture, in our architecture the information flows back from the lower to the higher resolution levels only through the deformations, as opposed to their heavier U-net style approach.
>
> **Can you comment on the significance of the improvement of the registration results?**
>
> Please refer to the global response for a detailed clarification on why we believe there is a clear improvement compared to the baselines.
>
> Regarding statistical significance, this is indicated in Tables 1 and 2 with an asterisk, and it was calculated using a permutation test (explained in Appendix K). This obviously does not yet guarantee ultimate clinical significance, but showing that would be clearly beyond the scope of our paper (or a single ICLR paper in more general).
>
> **The advantage of symmetry and inverse consistency over loss-based approaches is not clear, as all methods seem to provide symmetric and inverse consistent registrations apart from numerical accuracy. Can you comment on this?**
>
> While it may seem that cycle consistency loss is small for all of the methods, the improvement of two magnitudes by our method compared to the baselines is actually visually very significant. This is now well demonstrated in Figure 6 of Appendix H in the updated manuscript.
>
> As for inverse consistency, the performance of SYMNet is indeed very good, and we only claim to achieve similar performance in this metric. That is visually demonstrated in the new Figure 7 of Appendix H.

---

> ### Author Response · Authors · 2023-11-22
>
> We once more thank Reviewer eYpy for their questions and comments. We hope our clarifications about the novelty of our work, and why we believe our method brings a clear improvement compared to the baselines, have thoroughly answered the main questions of the reviewer. Please let us know if any additional clarifications would be useful for strengthening the initial positive perception of our article.

---

> > ### Comment · Reviewer_eYpy · 2023-11-22
> >
> > Thank you for your response to the comments raised. As I and the other reviewers have pointed out, there is a lot of prior work on exactly this topic and the discussion of the differences of the methods is still weak.

---

> ### Author Response · Authors · 2023-11-23
>
> We sincerely thank the reviewer for highlighting this aspect the would benefit from additional clarifications. To address this, we have written a new _Appendix G: Related work_, whose __length is approximately 2.5 pages__, which in detail explains the methods found in literature that are closest to our work or otherwise relevant to it. The Appendix G consists of four subsections _G.1 Classical registration methods_, _G.2 Deep Learning Methods (Earlier work)_, _G.3 Deep Learning Methods (Recent Methods Parallel To Our Work)_, and _G.4 Novel aspects of our method_.
>
> We hope that this new, thorough and detailed discussion clarifies the position of our work with respect to existing literature, highlights the differences, and makes it clear what the novel aspects of our method are compared to the alternatives.

---

### Official Review · Reviewer_2iyN · 2023-11-02

**Soundness:** 2 fair
**Presentation:** 2 fair
**Contribution:** 2 fair
**Rating:** 5
**Confidence:** 4

**Summary:**

This paper considers the deformable medical image registration with symmetric, inverse consistent, and topology-preserving properties, which is achieved by construction via a multi-resolution deep neural network. The proposed method is compared with three existing methods, i.e., SYMNet, VoxelMorph, and cLapIRN. The experimental results demonstrate the effectiveness of the proposed approach.

**Strengths:**

This paper works on an interesting research problem. It integrates existing strategies to achieve all the symmetric, inverse consistent, and topology-preserving properties in one network using an end-to-end training.

**Weaknesses:**

1. Insufficient study on related work. Unlike what is stated in the paper, the LDDMM method has been used in deep learning, such as DeepFlash[1], NODEO[2], R2Net[3], etc. Also, the inverse consistency by construction using multistep deep registration [4], which is quite close to this paper, is missing in the related work and experimental comparison.

[1] Wang and Zhang, DeepFLASH: An Efficient Network for Learning-based Medical Image Registration, CVPR 2020.
[2] Wu et al., Nodeo: A neural ordinary differential equation based optimization framework for deformable image registration, CVPR 2022.
[3] Joshi and Hong, R2Net: Efficient and flexible diffeomorphic image registration using Lipschitz continuous residual networks, Medical Image Analysis, 2023.
[4] Greer et al., Inverse Consistency by Construction for Multistep Deep Registration, MICCAI 2023.

2. Unclear presentation with unexplained statements. Such as, 1) "However, SYMNet does not guarantee symmetricity by construct", why? How to draw this conclusion? 2) Denoting the feature extraction network by h, how is this feature extraction network designed? Do we need to pretrain it or train together with the following network? 3) Squaring and scaling are not enough to guarantee the diffeomorphic property of deformations, we need an additional loss term or strategies to enforce the smoothness of the initial velocity fields first. However, this paper uses the same loss term on the smoothness of deformations in non-diff VoxelMorph, is it a reasonable choice?

3. Insufficient experimental results. This paper should compare the two most related works, i.e., [Iglesias 2023] that is mentioned in the introduction and the above [Greer 2023] that is missing in the paper, and some traditional methods, like SyN in ANTs and Symmetric LDDMM. Also, can you show more qualitative results to demonstrate the improvement of the proposed method and analyze the contribution of each design accordingly?

4. All the strategies used in this paper are what were used before, so, what are the challenges of this work? Is it necessary to have such a complicated network in practice?

**Questions:**

Please check the weakness section for the questions.

---

> ### Author Response · Authors · 2023-11-15
>
> **Insufficient study on related work. Unlike what is stated in the paper, the LDDMM method has been used in deep learning, such as DeepFlash[1], NODEO[2], R2Net[3], etc.**
>
> We do not claim in the paper that LDDMM has not been used in deep learning at all but instead that it has not been used much due to its computational cost. The methods you proposed, although interesting, do not qualify as examples of applying the diffeomorphic LDDMM in deep learning in our setup. In more detail:
>
> - DeepFlash does not involve optimization over diffeomorphisms. Instead, it uses initial velocity fields generated by a traditional LDDMM method as the optimization targets. Hence, it is not applicable to our unsupervised setting.
> - NODEO is a very interersting method but to mitigate computational cost it has to do significant modifications to the default LDDMM framework and as a result it actually loses the by construct diffeomorphic properties. To achieve invertibility they have to use a separate loss function on the Jacobian determinants (Equation 10).
> - R2Net, similarily to NODEO, has to make simplications to the theoretical LDDMM framework. As a result, it does not guarantee diffeomorphisms by construct, but instead applies a loss function on the Jacobian determinants to enforce that (Equation 10).

---

> ### Author Response · Authors · 2023-11-15
>
> **Also, the inverse consistency by construction using multistep deep registration [4], which is quite close to this paper, is missing in the related work and experimental comparison.**
>
> The paper by Greer et al. (2023) is an interesting work independent of ours. It was published after the submission deadline of ICLR, and hence it was not discussed in our paper. We now included it in the related work section of the revised version.

---

> ### Author Response · Authors · 2023-11-15
>
> **1) "However, SYMNet does not guarantee symmetricity by construct", why? How to draw this conclusion?**
>
> Our statement simply reflects the fact that, while its optimization target is symmetric, there is nothing in the SYMNet architecture that would directly enforce the predictions to be symmetric. That is also experimentally shown by the cycle consistency metric results, and in the new Figure 6 (Appendix H). We added a reference to the figure in the text.

---

> ### Author Response · Authors · 2023-11-15
>
> **2) Denoting the feature extraction network by h, how is this feature extraction network designed? Do we need to pretrain it or train together with the following network?**
>
> We explain the design in the beginning of Section 3.2: "in practice h is a ResNet (He et al., 2016) style convolutional network and features at each resolution are extracted sequentially from previous features." No pretraining is used.

---

> ### Author Response · Authors · 2023-11-15
>
> **3) Squaring and scaling are not enough to guarantee the diffeomorphic property of deformations, we need an additional loss term or strategies to enforce the smoothness of the initial velocity fields first. However, this paper uses the same loss term on the smoothness of deformations in non-diff VoxelMorph, is it a reasonable choice?**
>
> It is true that both our method and the non-diff VoxelMorph apply the same loss on deformations to enforce smoothness. However, even if VoxelMorph is not diffeomorphic, our method still is, and this is guaranteed by the architectural constraints (Theorem 3.3).

---

> ### Author Response · Authors · 2023-11-15
>
> **This paper should compare the two most related works, i.e., [Iglesias 2023] that is mentioned in the introduction and the above [Greer 2023] that is missing in the paper, and some traditional methods, like SyN in ANTs and Symmetric LDDMM.**
>
> Both of these works are very recent, parallel with, and independent of our work. According to the ICLR reviewer guide, an empirical comparison with such methods is not expected. We have nevertheless included our updated paper to include references to these (Iglesias was included already before). SyN, on the other hand, is extremely slow (ANTs implementation ~30 min per one registration), and it has already been compared against our strong baselines with in general worse performance (Balakrishnan et al., 2019; Mok & Chung,
> 2020a;b). Hence, including it here would be both challenging and not expected to bring additional value.

---

> ### Author Response · Authors · 2023-11-15
>
> **Also, can you show more qualitative results to demonstrate the improvement of the proposed method and analyze the contribution of each design accordingly?**
>
> We updated the paper by including Appendix H, which demonstrates the imporovements in deformation regularity (Figure 5) and cycle consistency (symmetricity) (Figure 6), and showcases similar performance to SYMNet in inverse consistency (Figure 7).
>
> In short, our symmetric formulation ensures symmetricity, and together with the topology preserving design of the deformation prediction networks $u^{(k)}$, it ensures also inverse consistency and cycle consistency. Deformation regularity is also ensured by the latter. The multi-resolution architecture is required to achieve good tissue overlap metric performance.

---

> ### Author Response · Authors · 2023-11-15
>
> **All the strategies used in this paper are what were used before, so, what are the challenges of this work? Is it necessary to have such a complicated network in practice?**
>
> Multiple works in the past have tried to enforce one or all of the properties (symmetricity, inverse consistency, topology preservation), either by a loss function or by construct. However, only classical iterative registration algorithms have enforced all of them by construct, see e.g. Ashburner (2007) or Avants et al. (2008). There papers are very highly cited (~8000 and ~5000 citations), and their main point is exactly proposing a method which fulfills the properties by construct. The challenge we set out to tackle was to develop the first such method in the deep learning setting.
>
> To summarize, our novel methodological innovations were the trick to make the method symmetric (Section 3.1) and the new implicit deformation inversion layer (with potential impact outside this subfield), which uses 5 times less memory, making it feasible to apply the multi-resolution architecture for the large volumetric data in image registration. The work also involves non-trivial mathematical work of obtaining the optimal invertibility bounds (Appendix E). We showed how to integrate these innovations with well-established architectural building blocks for a method whose overall performance exceeded that of the existing methods. (To clarify, our innovations could be integrated with simpler networks to ensure the desirable properties, but for sota performance the best available building blocks need to be adopted, hence the "complexity".)

---

> ### Author Response · Authors · 2023-11-15
>
> ## References
>
> - Arsigny, Vincent, et al. "A log-euclidean framework for statistics on diffeomorphisms." Medical Image Computing and Computer-Assisted Intervention–MICCAI (2006)
> - Mok, Tony CW, and Albert Chung. "Fast symmetric diffeomorphic image registration with convolutional neural networks." Proceedings of the IEEE/CVF conference on computer vision and pattern recognition. 2020.
> - Mok, Tony CW, and Albert CS Chung. "Large deformation diffeomorphic image registration with laplacian pyramid networks." Medical Image Computing and Computer Assisted Intervention–MICCAI 2020: 23rd International Conference, Lima, Peru, October 4–8, 2020, Proceedings, Part III 23. Springer International Publishing, 2020.
> - Mok, Tony CW, and Albert CS Chung. "Conditional deformable image registration with convolutional neural network." Medical Image Computing and Computer Assisted Intervention–MICCAI 2021: 24th International Conference, Strasbourg, France, September 27–October 1, 2021, Proceedings, Part IV 24. Springer International Publishing, 2021.
> - Hering, Alessa, et al. "Learn2Reg: comprehensive multi-task medical image registration challenge, dataset and evaluation in the era of deep learning." IEEE Transactions on Medical Imaging 42.3 (2022): 697-712.
> - Joshi, Ankita, and Yi Hong. "R2Net: Efficient and flexible diffeomorphic image registration using Lipschitz continuous residual networks." Medical Image Analysis 89 (2023): 102917.

---

> ### Author Response · Authors · 2023-11-22
>
> We again thank Reviewer 2iyN for their time and suggestions. We hope our responses have been useful in clarifying the concerns. Please let us know if there are any other questions and we are happy provide additional clarifications. If the reviewer finds our replies satisfactory, can they please consider updating their rating?

---

### Official Review · Reviewer_nQ7w · 2023-11-04

**Soundness:** 3 good
**Presentation:** 3 good
**Contribution:** 3 good
**Rating:** 5
**Confidence:** 4

**Summary:**

This paper is about a new deformable image registration method which is able to extract multi-resolution features that are symmetric, inverse consistent, and topology-preserving. The new framework is new and symmetric and inverse consistent by construct. Based on the deep equilibrium network framework, a new deformation inversion layer is proposed.

**Strengths:**

The paper is well written with good introduction and descriptions of symmetric, inverse consistent, and topology-preserving registration methods.

The proposed DL architecture is inverse consistent and symmetric by construct, rather than by using loss functions.

The use of deformation inversion layers seems interesting, based on the deep equilibrium network framework.

**Weaknesses:**

Although the new framework is interesting, as listed in Table 2, the accuracy improvement is not significant.

As shown in Table 3, the computation efficiency and memory usage have not improved.

**Questions:**

1.	In Figure 1, the images x_1 and x_2 are not defined previously and are unclear to me.

2.	Cannot find Figure 3.2 in Section 3.2

3.	Is the image registration framework diffeomorphic?

**Details Of Ethics Concerns:**

N.A.

---

> ### Author Response · Authors · 2023-11-15
>
> Thank you for your comments, and especially for acknowledging that the proposed deformation inversion layer is interesting, which we believe might even have a wider impact outside this particular subfield.
>
> **Although the new framework is interesting, as listed in Table 2, the accuracy improvement is not significant.**
>
> While it is true that the accuracy improvement based on the tissue overlap metrics alone is not very large (although statistically significant) compared to one of the baselines (cLapIRN), the significance lies in the fact that our method achieves the accuracy with significantly more regular deformations. We added a new figure to the appendix to demonstate that visually (Figure 5, Appendix H).
>
> For a more detailed response on our results, which we actually believe showcase a significant improvement compared to the baselines, please see the global response.
>
> ## Questions
>
> **In Figure 1, the images x_1 and x_2 are not defined previously and are unclear to me.**
>
> Thanks for spotting this typo. We fixed it in the updated version.
>
> **Cannot find Figure 3.2 in Section 3.2**
>
> The intended reference was Figure 2 but it was shown incorrectly due to a mistake in latex syntax. It has been fixed in the updated version.
>
> **Is the image registration framework diffeomorphic?**
>
> Yes, Theorem 3.3 proves that the architecture is topology preserving (invertible), and smoothness is guaranteed by the B-spline representation (Appendix I).

---

> ### Author Response · Authors · 2023-11-22
>
> We again thank the Reviewer nQ7w for their time and suggestions. Our joint response clarifies why we believe there is a clear improvement in the results compared to the baselines. Also, we've explained how the 5x memory saving in the deformation inversion layer is essential to apply the multi-resolution registration (please see our reply to eYpy). We cordially ask the reviewer to consider increasing their score, if they are satisfied with our replies. We are still happy to provide additional clarifications.

---

### Author Response · Authors · 2023-11-15

We sincerely thank all the reviewers for their effort, comments, and feedback, and for acknowleding that our paper addresses an important and interesting problem in medical imaging, is well-written, has interesting new components such as the novel deformation inversion layer, and enforces multiple desirable characteristics by construct.

## Experimental results

Since all the reviewers asked for clarifications about our experimental results, we aim to clarify this aspect here jointly. It is true that the improvement with respect to an individual metric, when compared with some individual baseline and considered in isolation of the rest of the results, might seem small (although often statistically significant). However, this does not take into account the complexity of image registration performance evaluation, where the perfect ground truth is essentially never available. Instead, the comparison should be based on a holistic evaluation, based on a comprehensive assessment across all metrics (Pluim et al., 2016; Rohlfing, 2011). In such a comparison, the performance of our method is better by a significant margin, as discussed in the results section.

We'll clarify with a concrete example. A quick assesment might conclude that SYMNet produces more regular deformations than our method since the percentage of folding voxels is on average only $1.5 \times 10^{-3}$ whereas for our method it is around $8.1 \times 10^{-3}$ (OASIS data). However, one has to take into account the following two points:
 1. The deformation regularization parameter $\lambda$ actually involves a trade-off between regularity of the deformation and the dice score (and HD95). Since our method is much better on dice score, by slightly increasing the regularization we could easily obtain a model with $\approx 0$ folding voxels but still significantly better dice score than SYMNet. Indeed, based on the validation set results in Table A1 in Appendix II, by setting $\lambda = 2.0$, our model would be superior on every metric compared to SYMNet.
 2. Both of these numbers are anyway extremely small compared to other methods, and their small difference can be seen as irrelevant (for visualization, see Figure 5 in Appendix H of the updated manuscript). Hence, these two methods outperform other methods on this metric (but unlike our method, SYMNet is clearly worse in other respects)

The example demonstrates how one should look at the overall performance due to the trade-offs involved, and that is exactly the reason why recent image registration papers always use multiple metrics assessing performance from different perspectives, to prevent the authors of respective papers from making their results look good simply by, e.g., setting very low deformation regularization. Rohlfing (2011) showed an extreme example with a perfect dice score but completely ridiculous and unrealistic deformations. Also, one should do pair-wise comparisons between methods instead of aiming to find a method that outperforms other methods on every single metric.

To summarize, our method has significantly better overall performance than SYMNet. Similarily, while our method has similar tissue overlap metric performance (or actually slightly better) than cLapIRN, it has significantly better deformation regularity and cycle consistency (see the new Figures 5 and 6 in Appendix H), rendering our method clearly better than cLapIRN in terms of overall performance. Finally, our method outperforms VoxelMorph on every metric with a significant margin.

## References

- Rohlfing, Torsten. "Image similarity and tissue overlaps as surrogates for image registration accuracy: widely used but unreliable." IEEE transactions on medical imaging 31.2 (2011): 153-163.

- Josien PW Pluim, Sascha EA Muenzing, Koen AJ Eppenhof, and Keelin Murphy. "The truth is hard to make: Validation of medical image registration". In 2016 23rd International Conference on Pattern Recognition (ICPR), pp. 2294–2300. IEEE, 2016.

---

### Meta-Review · Area_Chair_CRU2 · 2023-12-18

**Metareview:**

The paper proposes a method for learning-based image registration that possesses several inductive biases (e.g., symmetry, etc.) by construction. Results and comparisons are shown on two brain MR datasets. The initial reviews were quite mixed, with concerns about the positioning of this work wrt. existing literature, evaluation, and significance of results. Some of these concerns have, in the AC's point of view, been addressed in the rebuttal; however, other concerns remain. Particularly, the AC agrees with the reviewers that the positioning of the paper wrt. existing work is suboptimal at best, and hence, the reader is left to judge potential contributions on his/her own. Also, while it is indeed true that authors are not required to compare non-published arXiv works, it would be good practice to accommodate a discussion of these works (even if superficial) in the rebuttal. Given the current comments and the rebuttal, as well as the open issues, I am recommending rejection at this point, as the AC believes that this work needs a substantially better framing and positioning of its contributions (including working out in detail the differences to the state-of-the-art).

**Justification For Why Not Higher Score:**

The paper lacks a clear positioning relative to the state-of-the-art. Especially, when all expert reviewers are  unsure about the actual contributions (even if they are there), parts of the manuscript need to be re-written and clarified. In it's current form the manuscript is not ready for publication.

**Justification For Why Not Lower Score:**

N/A

---

### Decision · Program_Chairs · 2024-01-16

Reject